# A metabologenomics approach reveals the unexplored biosynthetic potential of bacteria isolated from an Amazon Conservation Unit

Ana Carolina Favacho Miranda de Oliveira,[1,2] Bruna Domingues Vieira,[3] Rafael de Felício,[3] Lucas da Silva e Silva,[2] Adonney Allan de Oliveira Veras,[1] Diego Assis das Graças,[1] Artur Silva,[1] Rafael Azevedo Baraúna,[1] Daniela Barretto Barbosa Trivella,[3] Maria Paula Cruz Schneider[1,2]

**ABSTRACT**   The Amazon, an important biodiversity hotspot, remains poorly explored in terms of its microbial diversity and biotechnological potential. The present study characterized the metabolic potential of Gram-positive strains of the Actinomycetes and Bacilli classes isolated from soil samples of an Amazon Conservation Unit. The sequencing of the 16S rRNA gene classified the strains ACT015, ACT016, and FIR094 within the genera *Streptomyces*, *Rhodococcus*, and *Brevibacillus*, respectively. Genome mining identified 33, 17, and 14 biosynthetic gene clusters (BGCs) in these strains, including pathways for the biosynthesis of antibiotic and antitumor agents. Additionally, 40 BGCs (62,5% of the total BGCs) were related to unknown metabolites. The OSMAC approach and untargeted metabolomics analysis revealed a plethora of metabolites under laboratory conditions, underscoring the untapped chemical diversity and biotechnological potential of these isolates. Our findings illustrated the efficacy of the metabologenomics approach in elucidating secondary metabolism and selecting BGCs with chemical novelty.

**IMPORTANCE**   The largest rainforest in the world is globally recognized for its biodiversity. However, until now, few studies have been conducted to prospect natural products from the Amazon microbiome. In this work, we isolated three free-living bacterial species from the microbiome of pristine soils and used two high-throughput technologies to reveal the vast unexplored repertoire of secondary metabolites produced by these microorganisms.

**KEYWORDS**   secondary metabolism, biosynthetic gene clusters, untargeted metabolomics, *Streptomyces*, *Rhodococcus*, *Brevibacillus*

The Amazon is the largest rainforest in the world and is considered an important source of biodiversity. The bioeconomy has been one of the strategic solutions for the social and economic development of the region. It recognizes biodiversity as an important commercial asset and, therefore, emphasizes the need for ecosystem conservation (1). The prospecting of natural products from Amazonian microbial communities has proven to be a promising strategy for discovering new molecules, including volatile organic compounds with antimicrobial activities and β-glucosidase inhibitors synthesized by cyanobacteria isolated from lagoons in the region (2, 3) and a new biosurfactant produced by Pseudomonas aeruginosa isolated from the soil of a mining area, which has antiviral, antitumor, and antimicrobial activities (4). Natural products (secondary metabolites, or specialized metabolites) are produced by ribosomal or non-ribosomal synthesis and play a crucial role in the adaptation of microorganisms to

Address correspondence to Rafael Azevedo Baraúna, rabarauna@ufpa.br.

The authors declare no conflict of interest.

their natural habitat (5, 6). Beyond their ecological importance, their chemically diverse structures have been extensively explored for the development of biotechnological products for a plethora of applications, including antibiotics for human health (7). Traditionally, most natural product discovery has been driven by top-down approaches, with bioactivity-guided fractionation of extracts from microbes and plants. However, the laborious steps of this kind of analysis and its technical limitations compromise the efficiency of discovery rates (8).

The omics technologies have enabled the increasing number of complete genomes deposited in public databases, which increased considerably since the emergence of high-throughput DNA sequencers approximately 20 years ago (9). Through genome mining techniques, it is possible to observe many Biosynthetic Gene Clusters (BGCs) that encode secondary metabolites with unknown functions (10). Additionally, advances in untargeted metabolomics due to increasingly sensitive tandem mass spectrometry (MS/MS) systems have allowed a more in-depth analysis of the metabolic components of interesting extracts (11). The integration of omics data has enhanced the efficacy of traditional drug discovery methods by facilitating access to biosynthetic pathways and the development of strategies to stimulate the activation of silent (cryptic) gene clusters in laboratory culture. This has enabled the identification and de-replication of known compounds, thereby facilitating the identification of new enzymatic pathways involved in the biosynthetic production of promising metabolites (12).

Metabologenomics (genome sequencing combined with mass spectrometry-based metabolomics) combined with the OSMAC strategy (One Strain Many Compounds) is a powerful multi-omics approach for discovering new secondary metabolites, being successfully employed in the discovery of novel alkaloids from *Lacinutrix shetlandiensis* isolated from the southern sea (13–15). The same strategy was used by Iacovelli and colleagues (16) to describe the sesquiterpene- and lactone-producing fungus *Anthostomella pinea*, including its enzymes with biotechnological relevance such as carbohydrate-active enzymes (CAZymes) and non-specific peroxygenases (UPOs) (16). This study shows the unexplored biosynthetic potential of three free-living bacterial strains isolated from soil samples collected at an environmental protection area near the city of Belém (PA, Brazil) at the Brazilian Amazon. The metabologenomic approach enabled the identification of the main classes of secondary metabolites produced by *Streptomyces* sp. ACT015, *Rhodococcus* sp. ACT016, and *Brevibacillus* sp. FIR094 and revealed a significant number of molecules that have not yet been explored.

## MATERIALS AND METHODS

### Sampling

Soil samples were collected from three points in the Utinga State Park, an environmental protection area near the city of Belém, Brazilian Amazon (Sample A: 1°25'35.6"S 48°25'28.2"W; Sample B: 1°25'30.6"S 48°25'35.0"W; Sample C: 1°25'28.3"S 48°25'42.9"W) (Fig. S1). Sampling was conducted during Amazon summer on October 7, 2021. A total of 10 g of soil was collected from a depth of 10 cm and stored at 4°C in previously sterilized polypropylene tubes.

### Bacterial isolation

Bacterial isolation was performed by serial dilution in saline solution (NaCl 0,9%) from 1 g of soil with subsequent inoculation of aliquots of up to $10^{-4}$ on Starch Casein Agar (SCA) by spread plate method. The medium was supplemented with 100 µg/mL$^{-1}$ cycloheximide to inhibit fungal growth. Plates were incubated at 28°C ± 2°C for up to 72 h. The suggestive actinomycete colonies (with a tough texture and a dry appearance and filaments with or without aerial mycelia) and bacillus colonies were isolated by strake plate method on SCA and stored in 20% glycerol at −80°C in Biological Engineering Laboratory biobank.

## Taxonomic classification

The strains were taxonomically classified based on the sequence of their 16S rRNA gene. Isolates were grown on Tryptic Soy Broth (TSB), and genomic DNA was extracted using DNeasy Blood and Tissue kit (Qiagen) according to the manufacturer's protocol. The Qubit Fluorometer (Thermo Fisher Scientific) was used for DNA quantification, and DNA integrity was evaluated on 1% agarose gel. The 16S rRNA gene was amplified using the universal primer pair 8F (5′-AGAGTTTGATCCTGGCTCAG-3′) and 1492R (5′-G GTTACCTTGTTACGACTT-3′). The PCR reaction was performed using the GoTaq Green Master Mix (Promega) with 10 ng of DNA and both primers at 200 nM. Amplicons were purified using ExoSAP PCR Product Cleanup Reagent (Thermo Fisher Scientific) and sequenced on ABI Prism 3500 Genetic Analyzer (Thermo Fisher Scientific) using the BigDye Terminator v3.1 kit. The sequences were analyzed in BioEdit v.7.2 and compared with the NCBI GenBank database using the web BLASTn available at https://ncbi.nlm.nih.gov/blast.

## Selection of strains

The criteria adopted to select strains with biosynthetic potential were (i) they should belong to the Actinomycetes and Bacilli classes, which are known to produce secondary metabolites of biotechnological interest and (ii) they should present inhibitory activity against control strains. Inhibitory activity was evaluated using the cross-streak method (17). Briefly, a single streak of the isolate was inoculated on Tryptic Soy Agar (TSA) and incubated at 28°C±2°C for up to 48 h. Subsequently, two control strains, *Staphylococcus aureus* ATCC 29213 and *Escherichia coli* ATCC 25922, were cross-streaked near the isolate streak. After incubation, the formation of inhibition zones near the growth of the isolate was considered an indication of bioactivity, allowing the selection of metabolically promising strains.

## Genome sequencing

Genome sequencing was performed using short- and long-read approaches to the three selected strains. Short-read sequencing was performed on Ion GeneStudio S5 Plus (Thermo Fisher Scientific) using the Ion 550 Chip kit. Long-read sequencing was performed on PromethION P2 Solo (Oxford Nanopore Technologies) using a R10.4.1 flow cell. Genome assembly was performed using SPAdes v.3.15.5 (18) and Flye v.2.9.2 (19) for short and long reads, respectively. Prokka v.1.14.5 (20) was used for genome annotation, and BUSCO (21) was used to evaluate genome completeness. The MUMmer v.4.0.0 systems were used to perform genome alignment against references deposited in GenBank, and CIRCOS v.0.69-9 (22) was used to construct circular maps. COG (Cluster of Orthologous Groups) assignment was performed in eggNOG-mapper v. 2.1.12 with the eggNOG database v. 5.0.2 database (23). Additionally, the Kyoto Encyclopedia of Genes and Genomes (KEGG) was used to obtain functional information of the predicted genes (24).

## Genome mining

AntiSMASH v.7.0.0 software (25) was used to identify BGCs in the sequenced genomes (22). The following parameters were used: detection strictness "relaxed" and "all on" extra features. Manual curation was performed after AntiSMASH prediction using the MIBiG platform (Minimum Information about a Biosynthetic Gene cluster) (26) and BLAST (27). Additional analyses were performed using GenBank (28), UniProt (29), Pfam (30), and the Protein Data Bank (PDB) (31).

## Sample preparation and metabolite extraction

The genomically characterized isolates were grown in different culture media to stimulate the expression of different classes of secondary metabolites (OSMAC strategy):

*Streptomyces* sp. ACT015 and *Rhodococcus* sp. ACT016 were grown in A1, TSB, TSBY, and ISP2 media while *Brevibacillus* sp. FIR094 was grown in LB, TSB, and R2A media. Erlenmeyer flasks containing 500 mL of medium were incubated at 30°C and 200 rpm for 48 h and subsequently sonicated in an ice bath for 10 cycles of 30 s at 30 s intervals. After lysis, the samples were centrifuged at 37,000 g for 30 min, and the supernatant was used for secondary metabolite extraction.

To 500 mL of culture, 400 mL (80% of the volume) of ethyl acetate was added in two stages (200 mL each stage). After the first separation of the organic phase from the aqueous phase, the aqueous phase was returned to the separation funnel, and the second portion of ethyl acetate was added. The organic phase portions were evaporated on a rotary evaporator at 40°C, 100 rpm. After complete solvent evaporation, they were resuspended with methanol (up to 10 mL) and transferred to pre-weighed 50 mL falcon tubes. The sample was dried using a SpeedVac (Thermo Scientific), and the solids were reweighed to calculate the obtained mass. For UPLC-MS/MS injection, the samples were resuspended in DMSO to a concentration of 10 mg mL$^{-1}$.

## Collection and analysis of data by UPLC-MS/MS

For the UPLC-MS/MS chemical profile, 2 µL aliquots from the samples were injected into a BEH C18 reversed-phase column (1.7 µm, 2.1 × 100 mm) coupled to a compatible pre-column, using a Waters Acquity H-Class UPLC system (Waters, Milford, MA, USA) coupled to a Bruker Impact II UHR-ESI-QqTOF mass spectrometer (Bruker Daltonics, Billerica, MA, USA). A solvent system of water (A), acetonitrile (B), 2% formic acid (C), and methanol (D) was used to compose the following analytical method: flow rate of 0.5 mL/min; column temperature 40°C; 0–1 min, 5% B; 1–6 min, 5% B to 35% B (curve 6); 6–10 min, 35% B to 95% B (curve 1); 10–12 min, 95% B; 12–15 min, 95% D (curve 1); and 15–18 min for column equilibration to the initial phase. C was kept constant at 5% (final concentration of 0.1%). The mass spectrometer was operated in positive ion mode, scanning mass in the range of 30–2,000 Da, with an acquisition rate of 8 Hz. End plate offset = 500 Volts (V); Vcap was 4,500 V; nebulizer was 4.0 bar; drying gas flow (N2) was 10 L/min; and drying gas temperature was 200°C. This was followed by an MS/MS scan for the most intense ions, with a cycle time of 1 s and an absolute threshold (per 1,000 sums) of 1,500 cts. For MS2, the mass-to-charge ratio (*m/z*) below 200 Da was excluded, and the "active exclusion" function was used. Each run was automatically calibrated using HCOONa (10 mM). The calibrated spectra were converted into mzXML files through Data Analysis 4.3 and included bio CompassXport tools (Bruker Daltonics, version 4.0.0.8).

## NP³ MS workflow data processing and analysis

Data processing and analysis were carried out using the NP³ Mass Spectrometry Workflow platform (32). The Spectra Similarity Molecular Networking (SSMN) was useful for organizing spectra in terms of similarity, providing an overview of metabolite distribution, and visualizing clusters of similar metabolites (possible analogs or compound families). The molecular network Ionization Variant Annotation Molecular Networking (IVAMN) was used as the basis to select and exclude nodes present in chromatographic blank samples and their immediate first neighbors. Then, the remaining nodes marked as protonated representatives (protonated_representative column equal to one in the node count table) were selected and filtered. The construction of IVAMN and SSMN in Cytoscape (33) was carried out following a Python script that automates some tasks in the software using the py2cytoscape library (https:// github.com/cytoscape/py2cytoscape). After these automated steps, nodes marked as culture media (bed), self-loop nodes, and nodes without connections were manually removed. The MS/MS chemical structure was performed using spectrum matches with the Global Natural Products Social Molecular Networking (GNPS) (34) and *In Silico* Spectral Databases of Natural Products (UNPD-ISDB) (35) databases. Annotations were

manually reviewed, and the related MS/MS clusters were grouped and named based on the chemical groups found in SSMN.

## RESULTS

### Selection of strains and general genomic characteristics

A total of 50 bacterial strains were isolated from Amazonian soils and maintained in the biobank of the Biological Engineering Laboratory (Belém, PA, Brazil). The strains ACT015, ACT016, and FIR094 were selected according to the criteria presented in the methods section and were classified by their 16S rRNA gene sequences in the genera *Streptomyces*, *Rhodococcus*, and *Brevibacillus*, respectively (Fig. 1a through c). *Streptomyces* sp. ACT015 presented a genome size of 8.6 Mb and 7,161 coding sequences (CDSs), whereas *Rhodococcus* sp. ACT016 and *Brevibacillus* sp. FIR094 presented genome sizes of 5.8 and 6.4 Mb, and 5,910 and 3,303 CDSs, respectively. General genomic characteristics are summarized in Table 1 and are in accordance with the average values for each genus. The genomes of *Streptomyces* sp. ACT015 and *Rhodococcus* sp. ACT016 were assembled using long-reads, whereas the *Brevibacillus* sp. FIR094 genome was assembled using short reads (Table 1).

The *Rhodococcus* sp. ACT016 genome was assembled into two contigs. The smaller contig (642,065 Kbp) showed very low similarity to the 13 *Rhodococcus* complete genomes used as reference (Fig. S2). Some genes found in this contig were related to plasmid functions such as the *tra* genes, which are directly involved in conjugative DNA transfer. One of these genes had a conjugative relaxase domain TrwC (Fig. S3a). Additionally, the plasmid-encoded enzyme AtzC was partially detected, presenting 97% similarity to the *Rhodococcus* sp. SGAir0479 gene (Fig. S3b). However, it was not possible to identify the plasmid incompatibility group using the PlasmidFinder v. 2.1.6 database (36).

Genomic rings were drawn to compare the genomes of ACT015, ACT017, and FIR094 against other reference strains with complete genomes deposited in GenBank (Fig. 2a, 3a, and 4a) (Table S1). CDSs containing COGs of unknown function (S) were the most represented in the three genomes (1,085 CDSs in *Streptomyces* sp. ACT015; 1,208 CDSs in *Rhodococcus* sp. ACT016; and 1,227 CDSs in *Brevibacillus* sp. FIR094), followed by transcription (K) (914 CDSs in *Streptomyces* sp. ACT015; 653 CDSs in *Rhodococcus* sp. ACT016; and 589 CDSs in *Brevibacillus* sp. FIR094). The transcription category encompasses activator or repressor proteins and sigma factors; therefore, they are key genes for environmental adaptation. The most abundant COG pathways were related to primary cellular functions such as carbohydrate metabolism, protein metabolism, and inorganic ion metabolism (Fig. 2b, 3b, and 4b). Other primary metabolic pathways were also

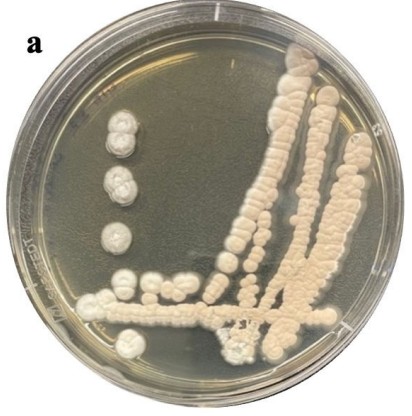
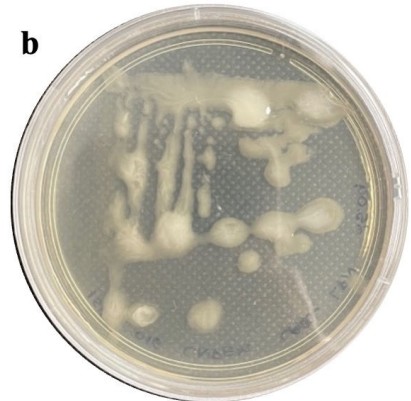
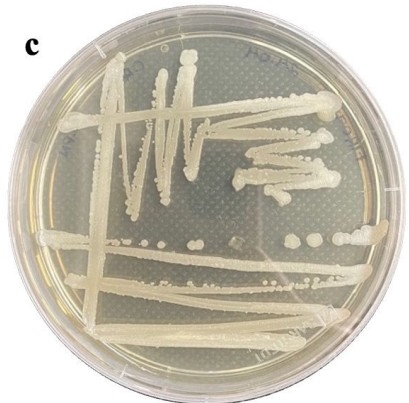

**FIG 1** Visual aspect of colonies of (a) *Streptomyces* sp. ACT015, (b) *Rhodococcus* sp. ACT016, and (c) *Brevibacillus* sp. FIR094, grown on Tryptone Soy Agar (TSA).

**TABLE 1** General genomic characteristics of the three isolated strains

| Strains | *Streptomyces* sp. ACT015 | *Rhodococcus* sp. ACT016 | *Brevibacillus* sp. FIR094 |
|---|---|---|---|
| Assembly | | | |
| Genome size (pb) | 8,643,132 | 5,868,135 | 6,348,900 |
| Number of contigs | 1 | 1 | 117 |
| N50 (bp) | 8,643,132 | 5,868,135 | 116,003 |
| G + C (%) | 73 | 68,4 | 47,2 |
| Coverage (x) | 329 | 82 | 507 |
| BUSCO | | | |
| Complete and single copy | 1,562 | 730 | 437 |
| Complete and duplicate | 6 | 11 | 4 |
| Fragmented | 6 | 2 | 6 |
| Missing | 5 | 0 | 3 |
| Total BUSCO genes | 1,579 | 743 | 450 |
| BUSCO completeness (%) | 99,4 | 99,8 | 98,0 |
| Annotation | | | |
| Number of CDS | 7,161 | 5,919 | 3,309 |
| Number of hypothetical genes | 1,579 | 743 | 2,333 |

detected such as those involved in carboxylic acid metabolism, nucleotide metabolism, and biosynthesis of amino acids and cofactors (Fig. 5).

The COG analysis identified 388 CDSs related to secondary metabolism in *Streptomyces* sp. ACT015, 422 in *Rhodococcus* sp. ACT016, and 178 in *Brevibacillus* sp. FIR094 (Fig. 2b, 3b and 4b). The KEEG enrichment, in contrast, found 370 CDSs related to secondary metabolism in *Streptomyces* sp. ACT015, 327 CDSs in *Rhodococcus* sp. ACT016, and 286

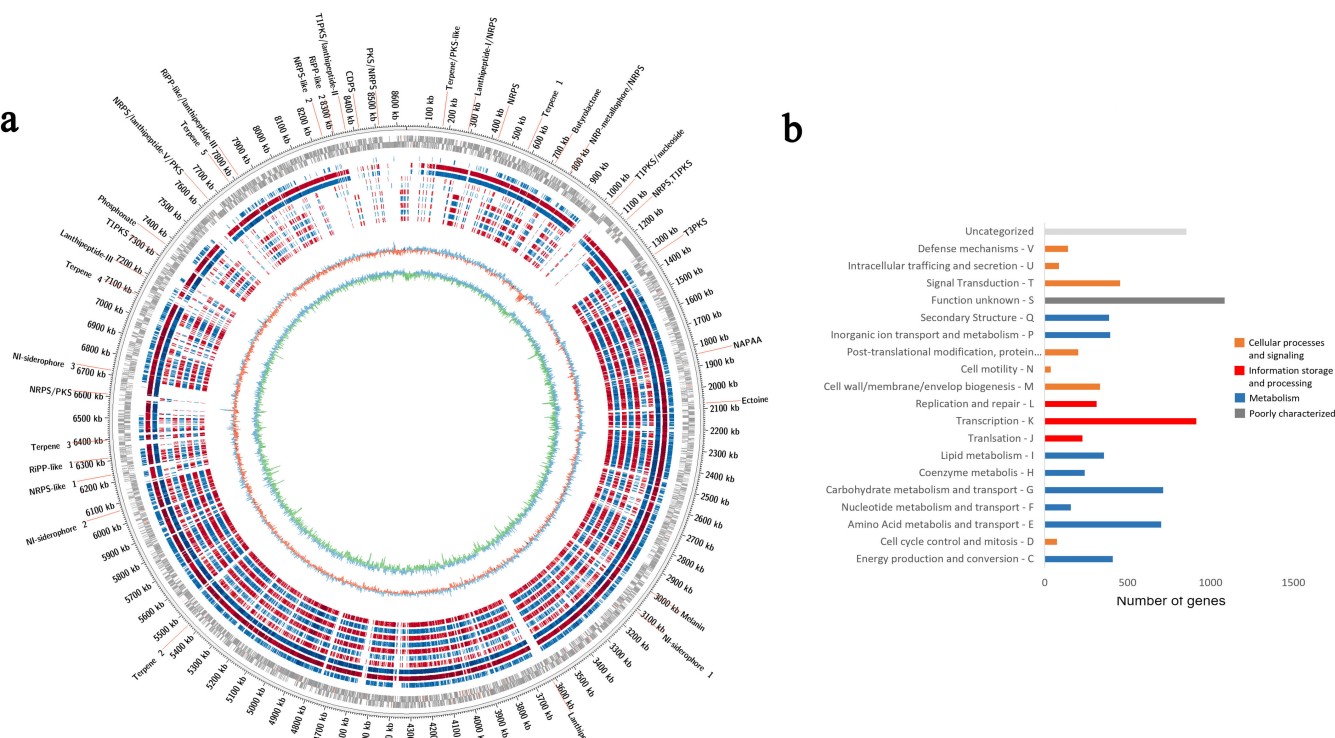

**FIG 2** (a) Circular map of *Streptomyces* sp. ACT015 genome. From the outermost ring to the innermost: locations of the 33 Biosynthetic Gene Clusters; genome size; Coding Sequences (forward and reverse); the genomes of *S. cadmiisoli* ZFG47; *S. griseoviridis* F1-27; *S. griseoviridis* K61; *S. koyangensis* SCSIO 5802; *S. koyangensis* VK-A60T; *Streptomyces* sp. HF10; *Streptomyces* sp. NEAU-sy36; *Streptomyces* sp. SUK 48; *Streptomyces* sp. WA1-19; and *Streptomyces* sp. WP-1; G + C skew; and G + C content. (b) Cluster of orthologous groups functional categories.

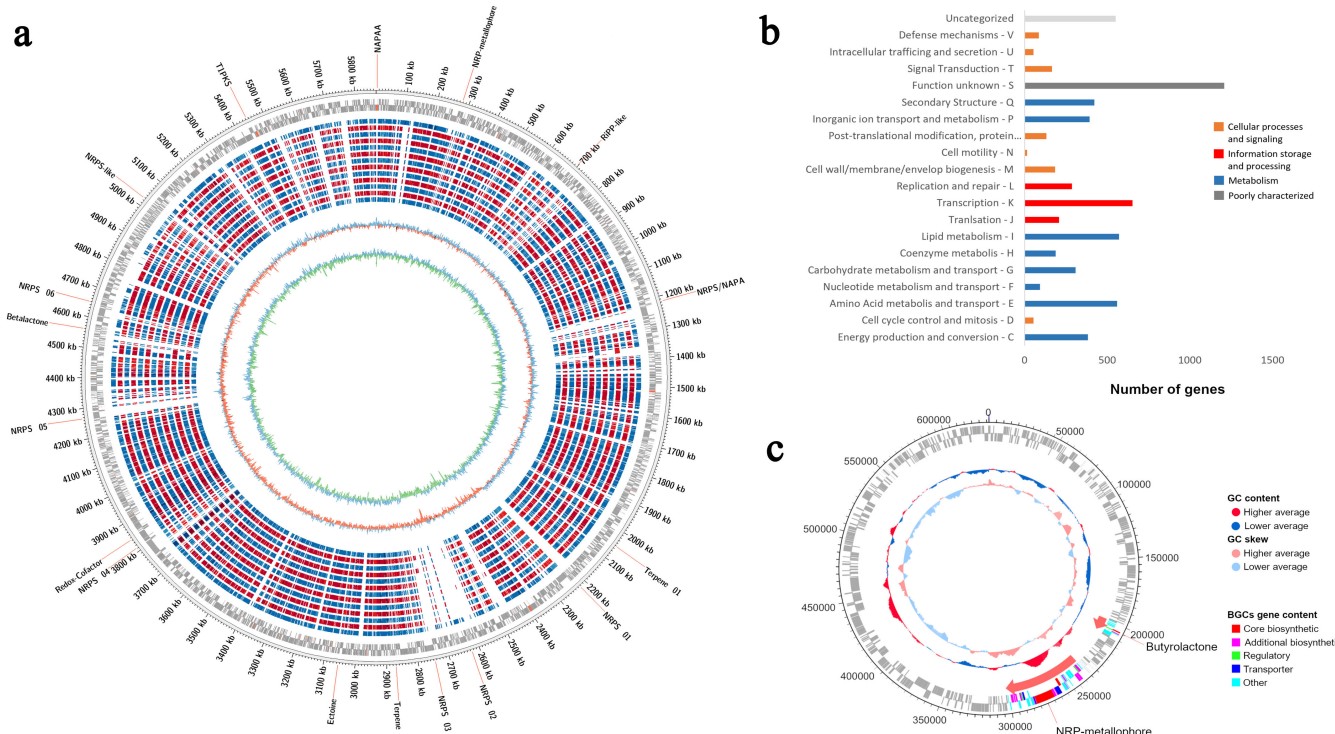

**FIG 3** (a) Circular map of *Rhodococcus* sp. ACT016 genome. From the outermost ring to the innermost: locations of the 17 biosynthetic gene clusters; genome size; coding sequences (forward and reverse); the genomes of *Rhodococcus* sp. W8901; *Prescottella equi* ATCC 33701; *P. equi* BJ13; *P. equi* DSSKP-R-001; *P. equi* FDAARGOS 952; *P. equi* JCM94-14; *P. equi* JCM94-27; *P. equi* JCM94-31; *P. equi* JCM94-3; *P. equi* JID03-46; *P. equi* JID03-56; *P. equi* PAM2287; and *P. equi* U19; G + C skew; and G + C content. (b) Cluster of orthologous groups functional categories; (c) Putative plasmid with 642,065 bp, 65% GC content and 2 BGCs.

in *Brevibacillus* sp. FIR094 and distributed in three heterogeneous profiles to the strains, especially for ACT015, the more external clade in Fig. 5. According to the genomic enrichment data (Fig. 5), the most abundant biosynthetic pathways in *Streptomyces* sp. ACT015 were related to the synthesis of terpenoids, polyketides, aminoglycosides, alkaloids, and beta lactams. *Rhodococcus* sp. ACT016 presented genes for the synthesis of terpenes (including carotenoids), siderophores, polyketides, aminoglycosides, and beta lactams, whereas *Brevibacillus* sp. FIR094 presented a significant number of pathways for the synthesis of terpenes, non-ribosomal peptides, and polyketides.

## Genome mining and biosynthetic potential

A total of 64 BGCs were identified in the genomes, most of which had low or no similarity to known clusters (62,5%). Occasionally, a BGC may contain one or more biosynthetic genes and therefore synthesize from the same cluster, producing different types of metabolites (37). This type of cluster was mainly observed in the *Streptomyces* sp. ACT015 and *Brevibacillus* sp. FIR094 genomes.

Thirty-three BGCs were detected in the genome of *Streptomyces* sp. ACT015. The main classes of biosynthetic genes were Polyketide Synthases (PKS), Ribosomally Synthesized and Post-translationally modified Peptide (RiPP), Non-Ribosomal Peptide Synthetase (NRPS), terpenes, and hybrid NRPS-PKS. Eighteen of 33 BGCs showed 70% homology with the MIBiG database, including BGCs responsible for the biosynthesis of carotenoids, mycelin, candicidin, flavolin, e-Polylysine, ectoin, melanin, deferoxamine, albaflavenone, indigoidine, minimycin, geosmin, antimycin, splenocin C, neoantimycin, trioxacarcin A, hopene, streptazone E, neocarziline A/B, SapB, birimostide, and ethylenediaminesuccine acid hydroxyarginine (EDHA). Among the 15 BGCs with low or no homology, 6 were related to RiPP, 4 NRPS, 4 PKS, 2 NRPS-PKS, and 2 terpenes (Table 2).

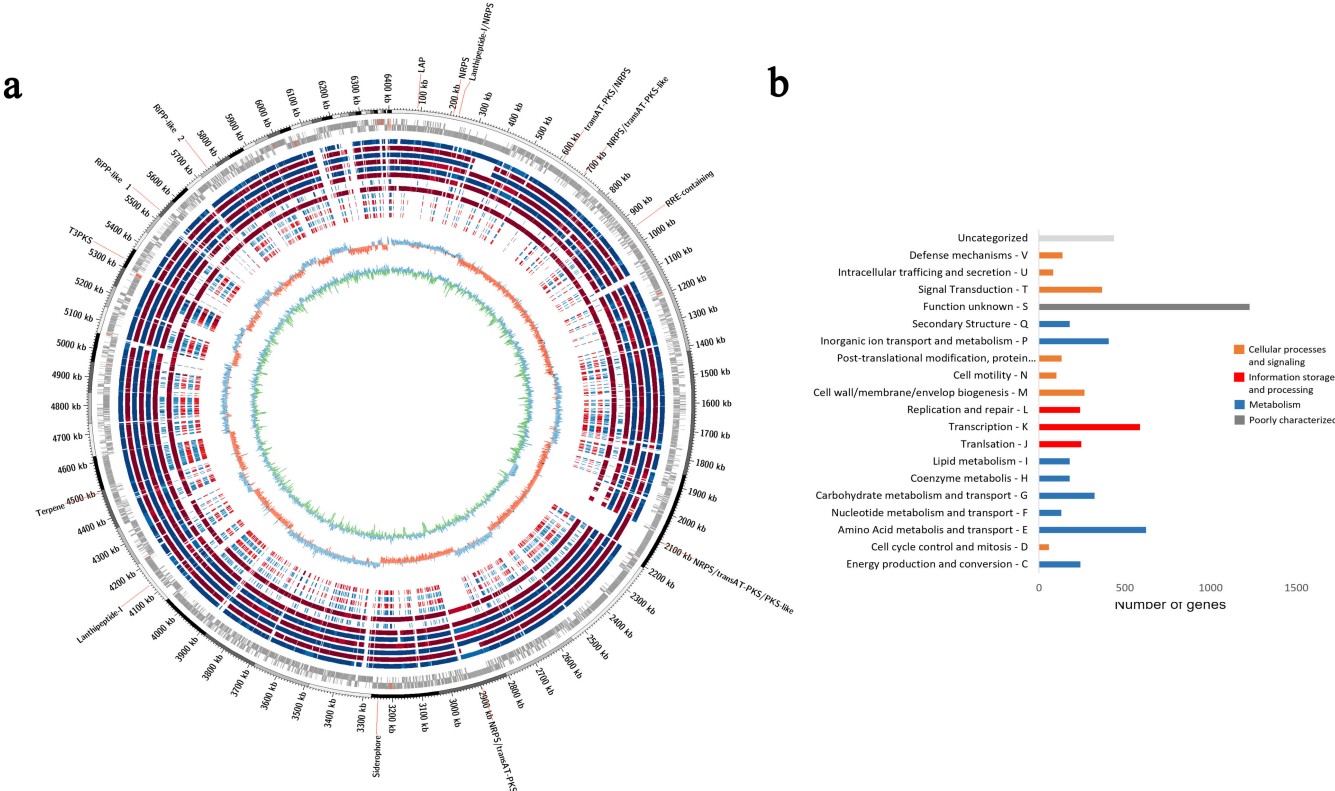

**FIG 4** (a) Circular map of *Brevibacillus* sp. FIR094 genome. From the outermost ring to the innermost: locations of the 14 biosynthetic gene clusters; genome size; coding sequences (forward and reverse); the genomes of *B. brevis* B011; *B. brevis* DZQ7; *B. brevis* HK544; *B. brevis* HNCS-1; *B. brevis* NBRC-100599; *B. brevis* NCTC2611-1; *B. choshinensis* HPD31-SP3; *B. formosus* NF2 ; *B. parabrevis* B3; *B. parabrevis* BCP-09; *Brevibacillus* sp. DP1.3A; and *Brevibacillus* sp. HD3.3A; G + C skew; and G + C content. (b) Cluster of orthologous groups functional categories.

In *Rhodococcus* sp. ACT016 genome, 17 BGCs were identified. The main classes were NRPS and terpenes. Only two had >70% similarity with MIBiG database and were predicted to be involved in the synthesis of e-Polylysine and ectoine. BGCs with similarity between 30% and 70% were related to the production of isorenieratene, corynecin, phosphoramidon, and thermochelin. The remaining 11 BGCs presented low or no similarity with the database, including 7 NRPS, 1 terpene, 1 RiPP, and 1 PKS (Table 3). Additionally, we were able to find two BGCs in the accessory contig, a putative plasmid, being an uncharacterized NRPS, and a Butyrolactone 63% similar to Heterobactin (Table 3).

Finally, 14 BGCs were identified in *Brevibacillus* sp. FIR094 genome. The main BGC classes were RiPP, NRPS, and hybrid NRPS-PKS. Only four showed >70% homology with the MiBiG database, including BGCs for the synthesis of tyrocidine, gramicidin, macrobrevin, and petrobactin. The other 10 BGCs showed low or no similarity, including 3 NRPS-PKS, 6 RiPPs, 1 PKS, and 1 terpene (Table 4).

## Untargeted metabolomics

To stimulate the production of as many secondary metabolites as possible, strains were grown in different culture media (32, 38). Molecular network analysis also facilitated the identification of variations caused by the OSMAC strategy (Fig. 6a). Considering all culture media, *Streptomyces* sp. ACT015 presented the richer raw molecular network containing 335 nodes, whereas *Brevibacillus* sp. FIR094 presented 312 nodes, and *Rhodococcus* sp. ACT016 presented only 100 nodes, totalling 747 nodes. The strains shared a few nodes, for example, *Streptomyces* sp. ACT015 and *Rhodococcus* sp. ACT016 shared only four nodes related to primary metabolism (Fig. 6b).

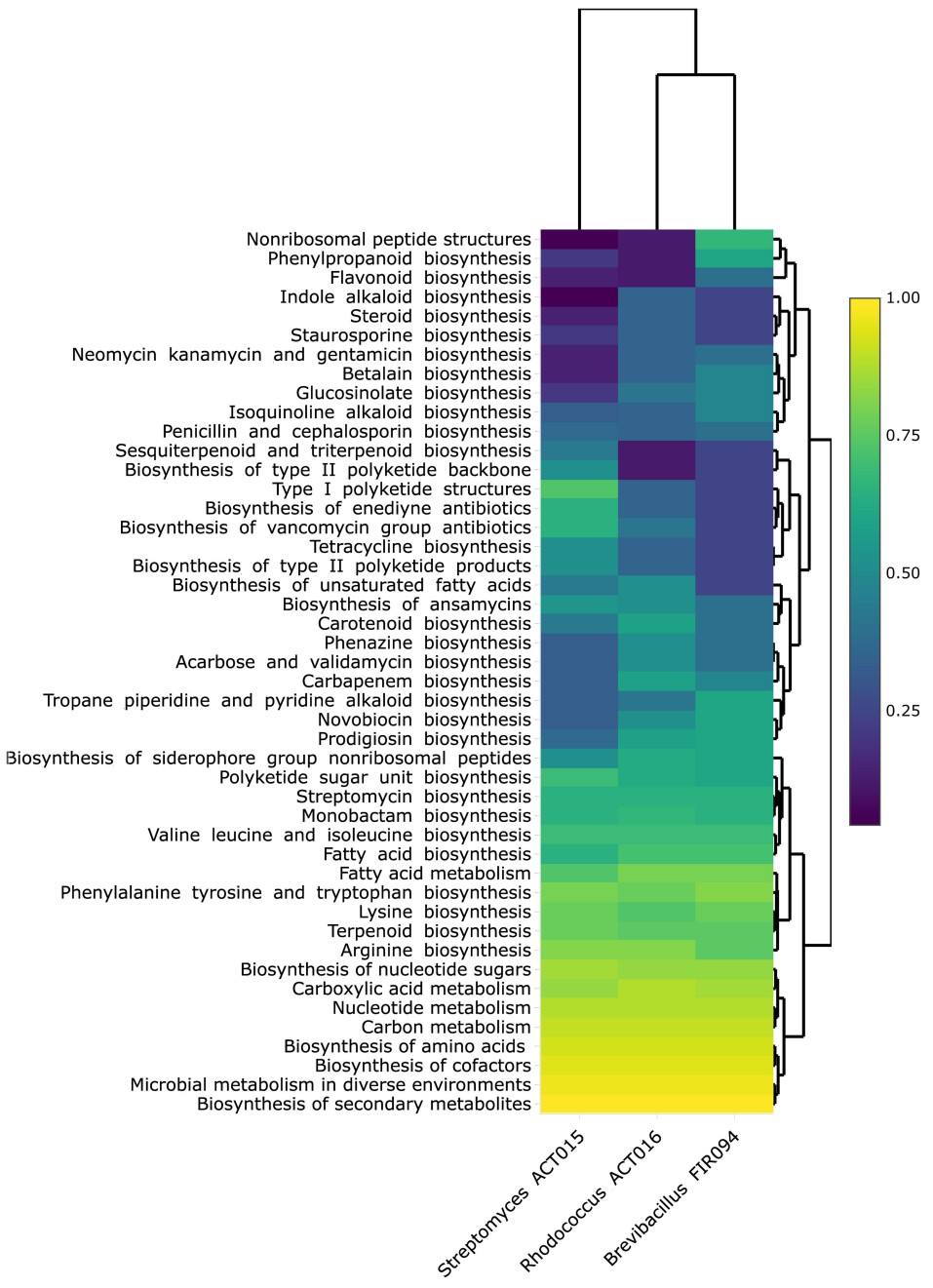

**FIG 5** Heatmap of enriched KEGG metabolism pathways. The color scale (right side) corresponds to the relative value of metabolites enriched in each KEGG pathway, being yellow the "highest" enrichment, and blue the "lower" enrichment. The upper dendrogram was generated using hierarchical clustering and represents the distance across the predicted metabolic profiles.

*Streptomyces* sp. ACT015 produced more metabolites in ISP2 and A1 culture media (Fig. 6c). In contrast, *Rhodococcus* sp. ACT016 produced more metabolites in A1 and TSB media (Fig. 6d), whereas R2A was the best medium for *Brevibacillus* sp. FIR 094 (Fig. 6e). Approximately 135, 40, and 119 nodes of *Streptomyces* sp. ACT015, *Rhodococcus* sp. ACT016 e *Brevibacillus* sp. FIR 094, respectively, were annotated by GNPS and/or UNPD-USDB databases. In contrast, 89, 23, and 105 nodes were not identified. When we applied restrictive parameters for spectra annotation (GNPS: Score >0.85; UNPS-ISDB

**TABLE 2** Biosynthetic gene clusters predicted by AntiSMASH in the *Streptomyces* sp. ACT015 genome

| BGC | Biosynthetic pathway | Similar cluster | Similarity |
|---|---|---|---|
| BGC 1 | Terpene | Isorenieratene | 100% |
| | | Carotenoid | 87% |
| | PKS | Rustimicin | 33% |
| BGC 2 | RiPP-Lanthipeptide I | - | 0% |
| | RiPP-Lanthipeptide II | | |
| | NRPS | | |
| BGC 3 | NRPS-PKS | Detoxin P1/P2/P3 | 50% |
| | | Detoxin N2/N3 | 40% |
| BGC 4 | Terpene | - | 0% |
| BGC 5 | Butyrolactone | - | 0% |
| BGC 6 | NRPS | Amychelin A/B | 81% |
| | | Cahuitamycin A/B/C | 62% |
| BGC 7 | T1PKS | Chlorothricin | 32% |
| | Nucleoside | - | 0% |
| BGC 8 | NRPS-T1PKS | Candicidin | 100% |
| BGC 9 | T3PKS | Flaviolin | 100% |
| BGC 10 | Other | ε-Poli-L-lysine | 100% |
| BGC 11 | Other | Ectoine | 100% |
| BGC 12 | Other | Melanin | 100% |
| BGC 13 | Other | Deferoxamine E | 100% |
| | | Deferoxamine A/B | 83% |
| | | FW0622 | 62% |
| BGC 14 | RiPP-Lanthipeptide | - | 0% |
| BGC 15 | Terpene | Albaflavenone | 100% |
| BGC 16 | PKS | Kinamycin | 13% |
| BGC 17 | NRPS | Indigoidine | 100% |
| | | Minimycin | 100% |
| BGC 18 | RiPP | - | 0% |
| BGC 19 | Terpene | Geosmin | 100% |
| BGC 20 | NRPS-PKS | Antimycin | 100% |
| | | Splenocin C | 93% |
| | | Neoantimycin | 76% |
| | PKS | Trioxacarcin A | 71% |
| BGC 21 | Siderophore | Paulomycin | 13% |
| BGC 22 | Terpene | Hopene | 92% |
| BGC 23 | RiPP-Lanthipeptide | - | 0% |
| BGC 24 | T1PKS | Streptazone E | 75% |
| BGC 25 | Phosphonate | - | 0% |
| BGC 26 | T1PKS | Neocarzilin A/B | 85% |
| | NRPS | - | 0% |
| | RiPP-Lanthipeptide V | - | 0% |
| BGC 27 | Terpene | - | 0% |
| BGC 28 | RiPP-Lanthipeptide III | SapB | 75% |
| BGC 29 | NRPS | Azalomycin | 13% |
| BGC 30 | RiPP | - | 0% |
| BGC 31 | RiPP-Lanthipeptide | Birimositide | 100% |
| | T1PKS | Amicomycin | 25% |
| | Betalactone | - | 0% |
| BGC 32 | Other | EDHA | 100% |
| BGC 33 | NRPS-T1PKS | Totopotensamide A/B | 58% |
| | T3PKS | - | 0% |

(*Continued on next page*)

**TABLE 2** Biosynthetic gene clusters predicted by AntiSMASH in the *Streptomyces* sp. ACT015 genome (*Continued*)

| BGC | Biosynthetic pathway | Similar cluster | Similarity |
|---|---|---|---|
| NRPS | - | 0% | |

*a*"-" in the "similar cluster" column means that no biosynthetic cluster similar to a known molecule present in the MiBIG databases was identified.

**TABLE 3** Biosynthetic gene clusters predicted by AntiSMASH in the *Rhodococcus* sp. ACT016 genome

| BGC | Biosynthetic pathway | Similar cluster | Similarity |
|---|---|---|---|
| BGC 1.1 | Other | ε-Poly-L-lysine | 100% |
| BGC 1.2 | NRPS | Thermochelin | 38% |
| BGC 1.3 | RiPP | - | 0% |
| BGC 1.4 | NRPS | Calicheamicin | 8% |
| BGC 1.5 | Terpene | - | 0% |
| BGC 1.6 | NRPS | - | 0% |
| BGC 1.7 | NRPS | - | 0% |
| BGC 1.8 | NRPS | - | 0% |
| BGC 1.9 | Terpene | Isorenieratene | 42% |
| BGC 1.10 | Other | Ectoine | 75% |
| BGC 1.11 | NRPS | Corynecin | 53% |
| BGC 1.12 | Redox cofactor | - | 0% |
| BGC 1.13 | NRPS | - | 0% |
| BGC 1.14 | Betalactone | - | 0% |
| BGC 1.15 | NRPS | - | 0% |
| BGC 1.16 | NRPS | - | 0% |
| | Other | Phosphoramidon | 40% |
| BGC 1.17 | T1PKS | - | 0% |
| BGC 2.1 | Butyrolactone | - | 0% |
| BGC 2.1 | NRPS | Heterobactin | 63% |

*a*"-" in the "similar cluster" column means that no biosynthetic cluster similar to a known molecule present in the MiBIG databases was identified.

Score >0.4; and Error <20 ppm), the number of identified spectra decreased drastically (Fig. 6f).

One hundred and twenty nodes were annotated in the *Streptomyces* sp. ACT015 metabolome, 85 of which were annotated using the UNPD-ISDB database and 50 using the GNPS database. Eighty-nine nodes were not annotated, corresponding to 42% of the metabolome. *Rhodococcus* sp. ACT016 produced 32 secondary metabolites that were annotated using UNPD-ISDB and GNPS databases, and 23 metabolites had no match (approximately 41% of the metabolome). *Brevibacillus* sp. FIR094 produced the largest number of unidentified metabolites (105 nodes), corresponding to 54% of the metabolome. Of the 88 secondary metabolites annotated, 41 had matches with the GNPS database and 78 were annotated with the UNPD-ISDB database.

Among the main superclass, the *Streptomyces* sp. ACT015 metabolome highlights the presence of organic acids, organic oxygen compounds, organ heterocyclic compounds, lipids, alkaloids, benzenoids, macrolides, peptides and oligopeptides, polyketides, and terpenoids (Fig. 7a). In the *Rhodococcus* sp. ACT016 metabolome the following superclasses were the most prominent: organic acids, organic oxygen compounds, organ heterocyclic compounds, lipids, alkaloids, and terpenoids (Fig. 7a). Finally, in the *Brevibacillus* sp., FIR094 metabolome showed mostly organ heterocyclic compounds, lipids, alkaloids, benzenoids, oligopeptides, polyketides, and terpenoids (Fig. 7a).

Some BGCs predicted by genome mining were possible to be confirmed with the metabolites annotated by LC-MS/MS. For example, BGC 20 (Table 2) from *Streptomyces* sp. ACT015 showed 100% similarity with the NRPS-PKS gene cluster that produces Antimycin (**compound 1**). This metabolite was annotated in its metabolome by using

TABLE 4 Biosynthetic gene clusters predicted by AntiSMASH in the *Brevibacillus* sp. FIR094 genome

| BGC | Biosynthetic pathway | Similar cluster | Similarity |
|---|---|---|---|
| BGC 1 | RiPP-LAP | Youssoufene | 11% |
| BGC 2 | NRPS | Tyrocidine | 81% |
| BGC 3 | NRPS | Gramicidin | 91% |
|  | RiPP-Lanthipeptide I | - | 0% |
| BGC 4 | NRPS-PKS | Zwittermicin A | 11% |
| BGC 5 | NRPS-PKS | - | 0% |
| BGC 6 | RiPP | - | 0% |
| BGC 7 | T1PKS Trans-AT | Macrobrevin | 100% |
|  | NRPS | - | 0% |
| BGC 8 | NRPS-PKS Trans-AT | Aurantinin | 25% |
| BGC 9 | Siderophore | Petrobactina | 83% |
| BGC 10 | RiPP | TVA-YJ-2 | 4% |
| BGC11 | Terpene | - | 0% |
| BGC 12 | PKS | Bacitracin | 22% |
| BGC 13 | RiPP | - | 0% |
| BGC 14 | RiPP | - | 0% |

*a*"-" in the "similar cluster" column means that no biosynthetic cluster similar to a known molecule present in the MiBIG databases was identified.

the GNPS database (**1**, *m/z* 506.3116) (Fig. 7b). Another example was the BGC 13 from *Streptomyces* sp. ACT015 presented 100% similarity with a gene cluster that produces Deferoxamine E (**compound 2**), a molecule that was further annotated in the metabolome of the bacteria grown on ISP2 medium (**2**, *m/z* 601.355) (Fig. 7b).

Despite correlations with this degree of specificity could not be made for other BGCs, some metabolites were annotated in the same chemical classes as some BGCs predicted in the genomes. For example, the ion with *m/z* 219.1741 in SSMN from *Streptomyces* sp. ACT015 metabolome was annotated as a sesquiterpene class and likely corresponds to the product of BGC 15, which presented 100% similarity to a gene cluster involved in the synthesis of the sesquiterpene Albaflavenone (218.33 g mol$^{-1}$). Comparisons between the genome and metabolome of *Brevibacillus* sp. FIR094 and *Rhodococcus* sp. ACT016 also revealed several correlations between BGCs and metabolites at the chemical class level. However, in most cases, no direct correlation could be made.

## DISCUSSION

### *Streptomyces* sp. ACT015 is a resource of known and unknown NPs genetically rich and metabolically active

Genomic parameters such as genome size, number of CDSs, and GC content were in accordance with what is expected for isolates of the genus *Streptomyces* (39). The genome mining of *Streptomyces* sp. ACT015 revealed a high diversity of biosynthetic gene clusters (BGCs), including polyketide synthases (PKS); non-ribosomal peptides (RiPPs) such as lanthipeptides, terpenes, non-ribosomal peptides, and polyketides (NRPS); and hybrid NRPS-PKS, which were identified as the most predominant. Belknap and colleagues (10) analyzed 1,110 *Streptomyces* genomes and identified these same BGCs as the most common of the genus. Additionally, the authors underscored the capacity of *Streptomyces* to harbor BGCs that synthesize molecules with antitumor activity (10). In *Streptomyces* ACT015, the coding regions responsible for the biosynthesis of the antimycin family of antimicrobials were identified (Table 2). As natural products typically address multiple targets to maximize their biological effects (40), the antiproliferative mechanisms of antimycin and their analogs have been described in cell lines of lung cancer, oral cancer, and cervical cancer. It has been postulated that the combination of antimycin with therapeutic agents may facilitate the overcoming of drug resistance and potential relapses (41–43). The presence of a cluster for the biosynthesis

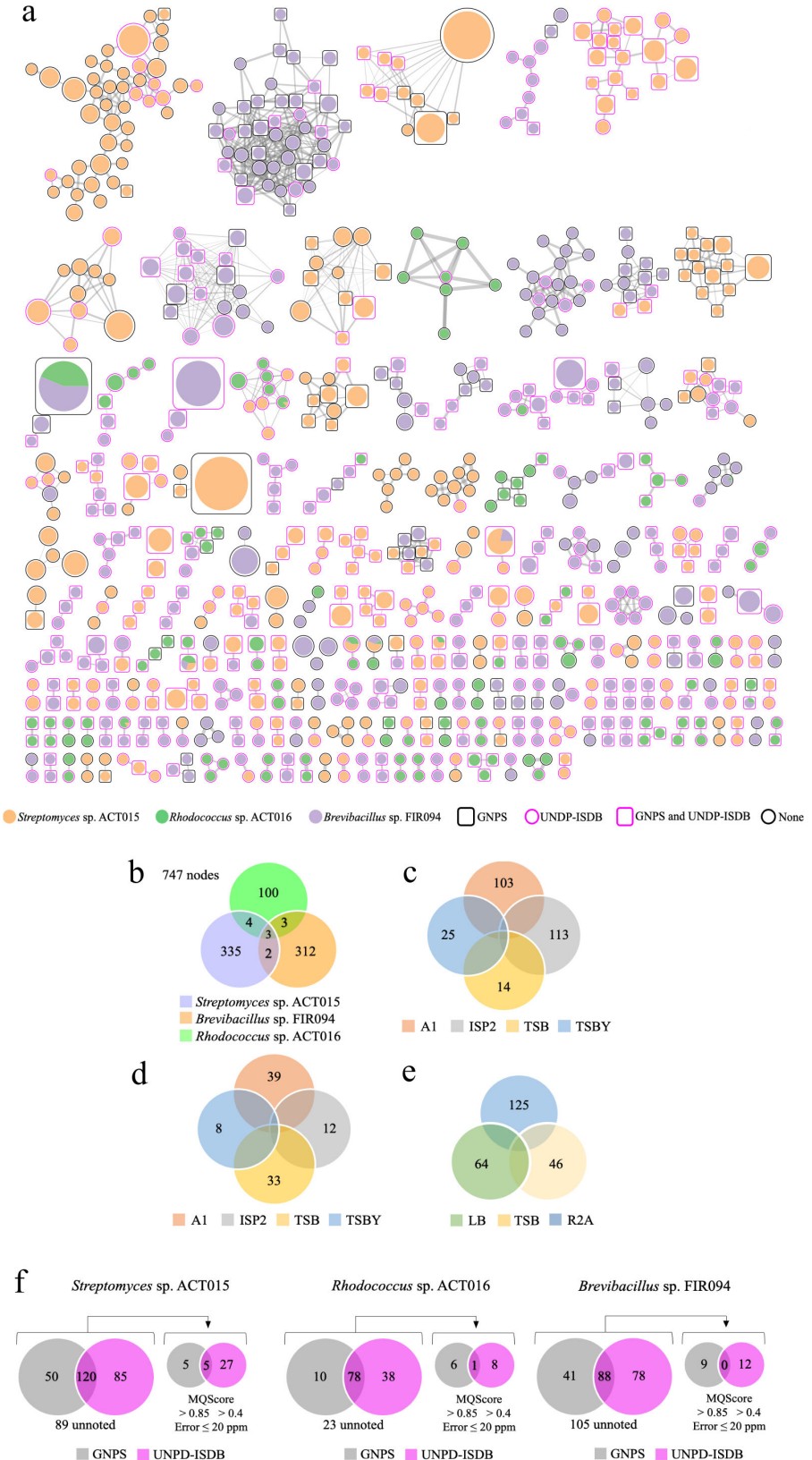

**FIG 6** (a) Spectra similarity molecular networking (SSMN) representing the secondary metabolites synthesized by the strains grown in different culture media. Each spectrum is represented by a node, and the level of similarity between nodes is directly proportional to the size of the edge that connects those nodes. Orange, purple, and green nodes represent spectra acquired

**Fig 6 (Continued)**

from the metabolome of *Brevibacillus* sp. FIR094, *Streptomyces* sp. ACT015, and *Rhodococcus* sp. ACT016, respectively. The size of the node indicates the percentage of ion contribution (peak area). Pink circles (UNPD-ISDB), black squares (GNPS), or pink squares (for both) identify the database responsible for the metabolite annotation. (b) Venn diagram with the total number of spectra detected in the crude extracts of the three strains. (c) Venn diagram illustrating the distribution of metabolite production in each of the culture media in *Streptomyces* sp. ACT015. (d) Venn diagram illustrating the distribution of metabolite production in each of the culture media in *Rhodococcus* sp. ACT016. (e) Venn diagram illustrating the distribution of metabolite production in each of the culture media in *Brevibacillus* sp. FIR094.

of neocarzilin also underscores the role of *Streptomyces* sp. ACT015 in the production of antitumor molecules (Table 2). Neocarzilin has been demonstrated to exhibit potent antimigratory and antiproliferative effects in both *in vitro* and *in vivo* breast cancer trials (44).

Given that the NRPS and PKS enzymes share the same assembly line organization, modules have been identified in ACT015 that house both pathways (BGCs 3, 8, 20, and 33 - Table 2). These clusters are commonly known as hybrid BGCs producing important specialized metabolites, such as salinosporamide A and eponemycin or TMC-86A, both potent proteasome inhibitors (45, 46). Although the origin and role of hybrid BGCs are poorly understood, these systems produce bioactive chimeric molecules that contribute significantly to the diversity of secondary metabolites observed in nature (47). The BGC 8 of the ACT015 strain contains the NRPS-T1PKS hybrid system, which is responsible for

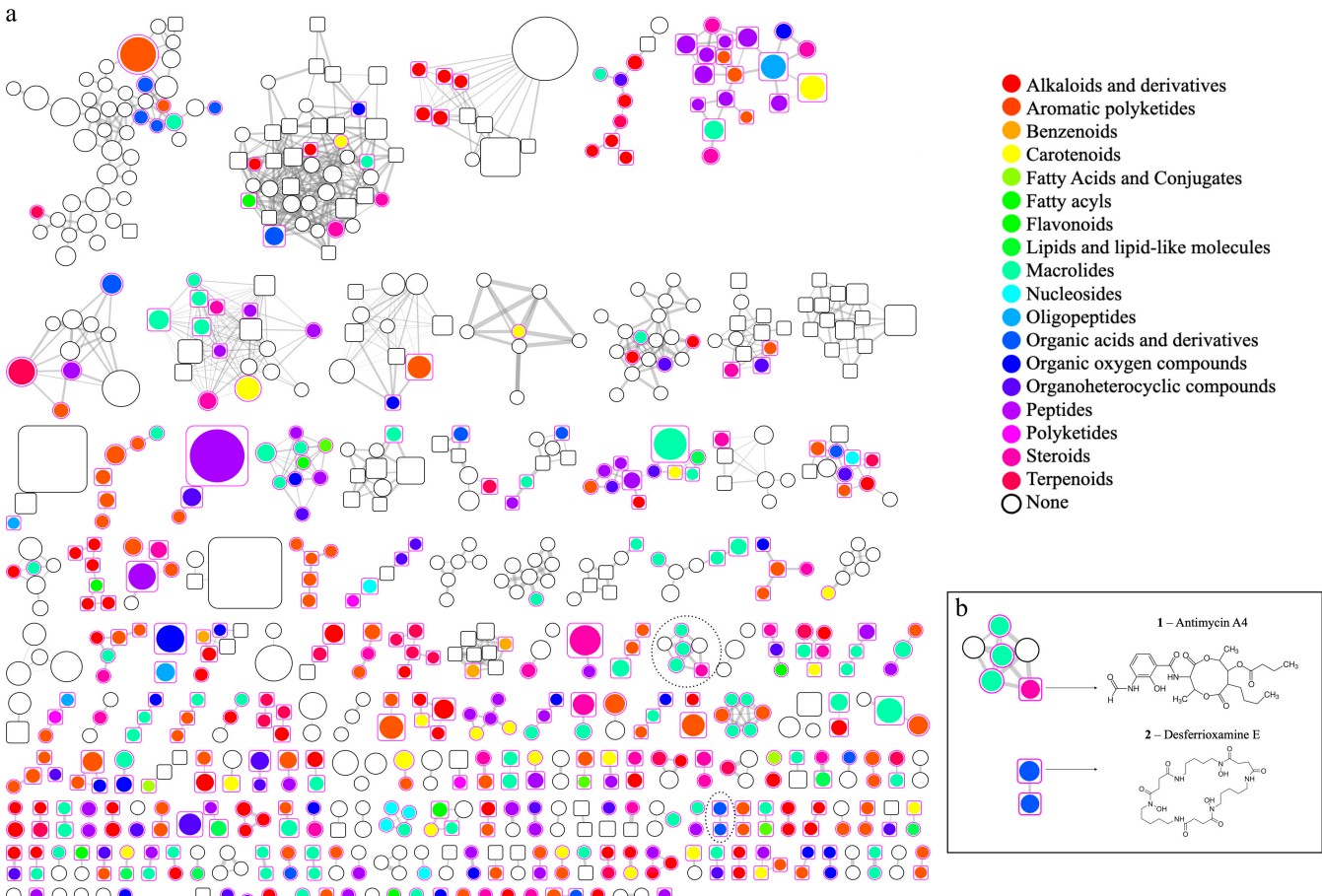

**Legend:**
- 🔴 Alkaloids and derivatives
- 🟠 Aromatic polyketides
- 🟠 Benzenoids
- 🟡 Carotenoids
- 🟢 Fatty Acids and Conjugates
- 🟢 Fatty acyls
- 🟢 Flavonoids
- 🟢 Lipids and lipid-like molecules
- 🟢 Macrolides
- 🔵 Nucleosides
- 🔵 Oligopeptides
- 🔵 Organic acids and derivatives
- 🔵 Organic oxygen compounds
- 🟣 Organoheterocyclic compounds
- 🟣 Peptides
- 🟣 Polyketides
- 🩷 Steroids
- 🔴 Terpenoids
- ⚪ None

**b**
1 – Antimycin A4
2 – Desferrioxamine E

**FIG 7** (a) SSMN represents the chemical superclass of each node. Each superclass is represented by a different color identified by the legend. Unidentified spectra are represented by white nodes. (b) LC-MS/MS analysis highlights the presence of Antimycin A4 (1) and Deferoxamine E (2) in *Streptomyces* sp. ACT015 crude extract, validating BGCs 20 and 13, respectively. The clusters are highlighted in the SSMN by dotted circles.

the biosynthesis of the antifungal agent candicidin. In this BGC, which is over 100 kb in size, three consecutive regulatory genes from the *luxR* family were identified as forming a regulatory subcluster. The regulatory proteins designated as PAS-LuxR have been identified in *Streptomyces* and have been shown to act in the transcription of functional polyene macrolide genes (including candicidin), aminoglycoside, and amino acid analogs (48). The PAS domain, a signaling module that monitors environmental changes such as light, redox potential, oxygen level, and small ligands, contributes to the increased complexity and precision of the *Streptomyces* regulatory system (49). The BGC 20 represents the most complex hybrid cluster of *Streptomyces* sp. ACT015, comprising three NRPS modules and four PKS modules, including those responsible for the antimycin family as previously discussed. In addition to the biosynthesis of antimycin, this BGC contains modules for the biosynthesis of splenocin, which has been demonstrated to be a potent inhibitor of pro-inflammatory cytokines. Its efficacy is comparable with that of the corticosteroid dexamethasone in a cytokine assay of splenocytes (50). Furthermore, it contains genes for the biosynthesis of trioxacarcine, a new class of DNA-modifying natural products with potent anti-proliferative effects (51).

Most of the BGCs in ACT015 genome had some similarity to previously described NPs (Table 2). Comparative genomics identified 5 BGCs of *Streptomyces* sp. ACT015 appears to be unique to the strain compared with the genomes used as reference genomes (Fig. 2a). BGC 7, predicted as T1PKS, was detected in a genomic region with a low level of similarity compared with the other genomes of the reference strains (Fig. 2a). Additionally, the AntiSMASH result indicates a low level of homology (25%) with a cluster from *Streptomyces* sp. CS081A. Another example was BGC 14 (Table 2; Fig. 2a). Despite this BGC shows 88% homology with *S. purpureus* JCM 3172 in AntiSMASH, it is responsible for synthesizing a molecule with no similarity to the MIBiG database. This analysis highlights the importance of comparative genomics studies to identify new potential secondary metabolites unique to a particular strain. The discovery of rare or exclusive BGCs indicates the existence of underexplored biosynthetic pathways that could lead to the production of new bioactive compounds with pharmaceutical and biotechnological applications.

Regarding the chemical classes of metabolites, *Streptomyces* sp. ACT015 produced essential metabolites for environmental adaptation such as carotenoids, melanin, geosmin, ectoin, hopene, and the siderophores amychelin and deferoxamines. These last two molecules are widely used in medicine. Amychelin was able to reduce the pathogenicity of *Pseudomonas aeruginosa* under infectious conditions (52). Deferoxamine was used in the treatment of iron overload, such as in beta-thalassemia disease (53) and cases of iron or aluminum poisoning (54). Other BGCs synthesized molecules with antibiotic potential, such as albaflavenone, indigoidine, minimycin, trioxacarcin A, and birimositide (55–59); antifungals such as candicidin (60); and chemotherapeutics such as minimycin, antimycin, neoantimycin and neocarzilin (44, 61, 62).

The considerable number of BGCs with unknown functions (15 of 33) and unannotated metabolic network nodes draws attention to a still unexplored biosynthetic potential. Despite *Streptomyces* being the taxon with the greatest number of entries in the MIBiG database (approximately 800), 15 BGCs of the ACT015 strain showed low or no similarity with the available data. Among these uncharacterized BGCs are 6 RiPPs. Due to the considerable variability in the sequences of precursor peptides and the catalytic reactions mediated by modification enzymes, genomic mining of RiPPs is promising for the discovery of new bioactive molecules (63). For example, the compound legonaridin, characterized by *Streptomyces* sp. CT34 and subsequently classified as a class B linaridin, has inspired genomic mining efforts for other class B linaridin with unique gene structures and significantly different precursor sequences (64, 65). Similarly, cluster analyses of kintamdin, a RiPP compound detected in *Streptomyces* sp. RK44 isolated from a soil sample, identified additional members of the β-amino acid family previously uncharacterized in RiPPs (66).

Although non-ribosomal peptides and polyketides are the major secondary metabolites of the genus *Streptomyces* (67), we found four uncharacterized BGCs for NRPS and four for PKS in ACT015. It is now possible to exploit this unknown genetic potential by cloning whole BGCs into heterologous expression vectors (68). Methods based on CRISPR-Cas and Cre-lox (CAPTURE) allow us to obtain natural products by capturing the biosynthetic region of interest in the wild organism (69). The application of this method allowed the successful cloning of 43 uncharacterized BGCs (10–113 kb in size) from 14 *Streptomyces* species and three *Bacillus* species. Subsequent heterologous expression yielded 15 new natural products, demonstrating that the use of this method is highly efficient and suitable for the discovery of new compounds (70). *Streptomyces* sp. ACT015 carries new gene clusters with unknown functions potentially involved in adaptation to the Amazon environment, a biome with a rich biodiversity. These BGCs can be used in future studies to characterize new natural products through genetic engineering and heterologous expression. The largest number of secondary metabolites were identified in ISP2 and A1 media. Bacterial growth and metabolism are directly impacted by the carbon source (71). ISP2 is routinely used for *Streptomyces* growth and antimicrobial activity assays (72). It contains glucose as a carbon source, whereas A1 medium contains soluble starch and TSBY, sucrose. The last medium used was TSB, which is composed of peptone digest and other components of lower concentration. The carbon catabolite repression (CCR) is a mechanism highly conserved in Streptomyces that regulates carbon metabolism. CCR also influences cell morphology and synthesis of some secondary metabolites (73).

Approximately 27% of the nodes in the *Streptomyces* sp. ACT015 metabolic network were not annotated. Identification of these metabolites is limited by the availability of reference spectra (50). The GNPS platform provides a large library of natural compounds for similarity analysis (34). However, annotation of small molecule mass spectra remains a challenging problem. Therefore, *in silico* tools with a larger spectral database can be used to complement annotation. We used ISDB (35) for *in silico* annotation of spectra. In fact, for the *Streptomyces* sp. ACT015 metabolome, the UNPD-ISDB annotated a greater number of spectra (85 nodes) compared with the GNPS (50 nodes). The tool confirmed the annotation of Antimycin A4 and Deferoxamine E as well as classified several other spectra into alkaloids, oligopeptides, peptides, polyketides, and terpenes. Desferoxamine E, also known as nocardamine, was annotated into oligopeptides superclass (Fig. 7b). This molecule is a cyclic hydroxamic acid siderophore with antitumor activity that is produced by several species of bacteria (74). Amino acids in cyclic conformation in the molecule structure may have induced its classification into the oligopeptide's superclass.

Interestingly, all clusters of the molecular network containing annotated compounds also contained unidentified adduct ion nodes that possibly have a similar chemical structure (Fig. 7a). These unidentified nodes may represent different adducts of the same known compound may characterize potential new analogs of predicted compounds. *In silico* tools are generally more accurate in predicting molecular classes, and therefore, special attention should be taken for more specific identifications. In this work, annotations were based on standard reliability parameters available in the molecular networks (GNPS: Score >0.85; UNPD-ISDB Score >0.4 e Error <20 ppm). Many annotations based on these criteria were discarded to reduce false positive annotations or were limited to the identification of the molecular class. Exploratory studies can take advantage of these *in silico* tools to better describe the biosynthetic potential of promising isolates.

## *Rhodococcus* sp. ACT016 is an environmental strain genetically rich and active with a highly shapable metabolism

The genome size of *Rhodococcus* species varies between 4 and 10 Mb. This large difference in genome sizes is related to the presence of large plasmids, extensive genome instability, and the diversity of habitats that are colonized by *Rhodococcus* species (75). Environmental strains such as *R. opacus* B4 usually have larger genomes

(8.2 Mbp) compared with pathogenic strains such as *R. equi* (5.4 Mbp) (76, 77). Some exceptions have been observed, such as *Rhodococcus* sp. Br-6 (5.5 Mbp) isolated from soil samples and *Rhodococcus defluvii* Ca-11 (5.1 Mbp) isolated from a wastewater treatment bioreactor (78, 79). The genome size of the strain ACT016 was 5.8 Mb, which is smaller than that expected for other environmental strains.

Although approximately 900 *Rhodococcus* genomes have been deposited at the GenBank, there is a lack of knowledge about the BGCs of this genus. Genome mining of 20 strains revealed an extensive repertoire of uncharacterized BGCs (80). In another study, Doroghazi and Metcalf (81) demonstrated that *Rhodococcus* isolates present a greater proportion of NRPS than actinomycetes. Interestingly, this was the main type of cluster found in the ACT016 genome, which possesses 17 BGCs, with 9 of them being NRPS. Additionally, 88% of the detected BGCs were not like BGCs previously described (Table 3). Some ACT016 clusters found as absent in other *Rhodococcus* reference genomes suggest this strain as harboring exclusive BGCs: the clusters 4, 6, 7, 8, and 13 (Table 3; Fig. 3), which were of particular interest because they were all NRPS. Among the BGCs identified using the AntiSMASH database, those with the highest identity were involved in the synthesis of ε-Polylysine, an antibiotic commonly produced by *Streptomyces* and *Kitasatospora*; and ectoine, a metabolite related to osmoprotection and survival in saline environments (82).

According to the untargeted metabolomics results, *Rhodococcus* sp. ACT016 showed a higher production of metabolites in A1 and TSB media, despite the lower node number when compared with ACT015, which was grown in the same culture media. The same metabolite was rarely synthesized in more than one culture medium, showing that the OSMAC approach was a success. Other metabolites from the classes of heterocyclic compounds, organic acids, lipids, terpenoids, polyketides, and alkaloids were also detected. Using the same approach as our study, Gamaleldin and colleagues (83) identified molecules with potential antimalarial activity produced by *Rhodococcus* strains. Palma-Esposito and colleagues (84) identified glycopeptides with potential antiviral activity.

Eleven nodes (10%) were related to lipid production, with no significant annotation. Some *Rhodococcus* species such as *Rhodococcus opacus*, *Rhodococcus jostii*, and *Rhodococcus wratislaviensis* are known to be oleaginous microorganisms, as they are capable of accumulating triacylglycerols (TAG) at more than 20% of their cell dry weight. Oleaginous rhodococci are a source of lipids with industrial application (85). High TAG production is obtained through growth under nitrogen starvation conditions or the abundance of carbon sources. An imbalance between carbon and nitrogen availability redirects carbon metabolism toward lipogenesis (86). Half of the nodes identified as lipids were produced in TSBY, which is composed of 10% sucrose and 3% soybean digest. The other half of the nodes were produced in A1 medium, which contains 1% soluble starch and 0.2% casein peptone, confirming that lipid production is strongly influenced by the carbon/nitrogen availability ratio.

### *Brevibacillus* sp. FIR094 is a metabolically active strain with conserved and still unknown NPs to be explored

*Brevibacillus* sp. FIR094 presented a genome with a size of 6.5 Mb and 14 BGCs. Most of these BGCs are widely distributed and conserved in the genus that synthesizes molecules with antimicrobial activity, including tyrocidine, gramicidin, and macrobrevin, but only four showed >70% similarity to the AntiSMASH database, suggesting an underexplored potential in FIR094 close-related strains.

Prospecting bacteria of biotechnological interest in underexplored environments is essential for the discovery of new molecules and to expand our understanding of the ecological context of natural antibiotic production. In *Brevibacillus* sp. Leaf182, which was isolated from an *Arabidopsis thaliana* leaf, several natural antibiotics, including a new trans-AT polyketide from the macrobrevin class, have been described (87). Macrobrevin analogs with antimicrobial activity were also identified by Chakraborty and colleagues

(88) in *Bacillus amyloliquefaciens* isolated from macroalgae. These two macrobrevin molecules presented only 46% similarity, demonstrating the genetic complexity involved in the production of similar compounds by different isolates.

*Brevibacillus* is a genus widely recognized as an Antimicrobial Peptide (AMP) producer, especially for AMPs synthesized by NRPS clusters. They also synthesize AMPs from RiPPs, especially bacteriocins, such as laterosporulin (89), laterosporulin10 (90), and Bac-GM100 (91). In *Brevibacillus* sp. FIR094, 6 RiPPs were detected, four of which showed low identity with known BGCs (Table 4). BGC 1 synthesizes a metabolite from the subfamily of linear azol(in)e-containing peptides (LAPs). LAPs comprise a subclass of RiPPs that share heterocyclic thiazol(in)e and (methyl)oxazol(in)e structures, promoting the cyclization of cysteine, serine, or threonine side chains during the post-translational modification step (92). This BGC was composed of the precursor peptide (gene A), the enzymes dehydrogenase (gene B) and cyclodehydratase (gene C/D), the maturation protein (gene F), and regulatory and transport genes (Fig. 8a). A metabolite of approximately 5,000 Da was calculated based on the 54 amino acid sequence of the core peptide. The peptide contains 14 cysteines (C) in a sequence that can form seven disulfide bonds (S-S). These disulfide bonds play a crucial role in the stabilization of the molecule, giving it a specific conformation and increasing its resistance to environmental factors. In addition, the sequence also included aromatic amino acids such as phenylalanine (F), which may contribute to the structural and functional properties of the peptide. The BGC also contained the transcription factor AbrC3, an inducer of antibiotic production (Fig. 9a). AbrC3 is a positive transcriptional regulator of specialized metabolites and morphological differentiation in *S. coelicolor* (93). The BGC of *Brevibacillus* sp. FIR 094 showed high similarity to other BGCs from the antiSMASH database, suggesting that the region is highly conserved within the genus (Fig. 9b). However, the metabolic

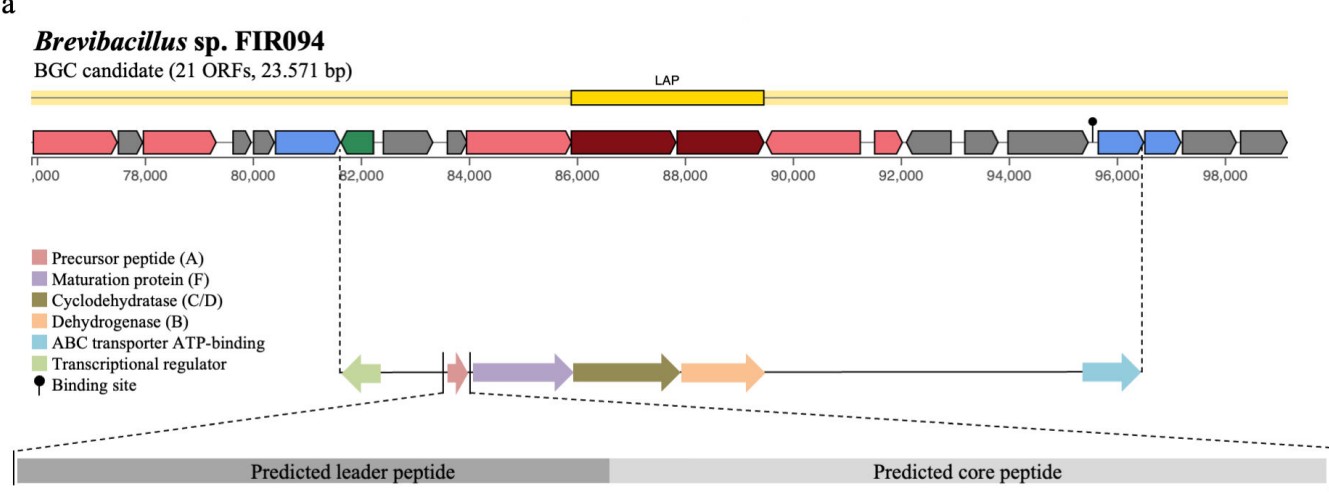

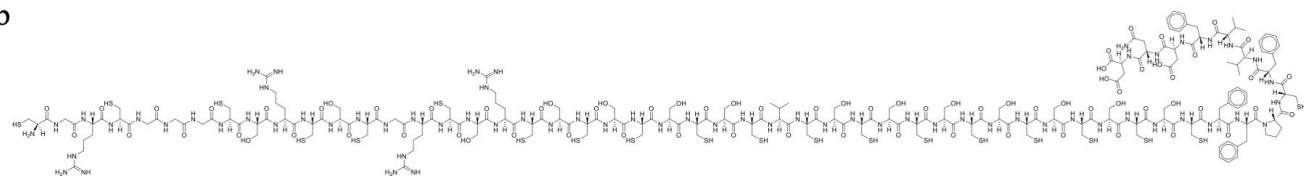

**FIG 8** (a) BGC from *Brevibacillus* sp. FIR094 suggested as a transcriptional region for RiPP subclass of linear peptides containing azole(s) (LAPs). The essential genes of the BGCs are depicted as colored arrows, including the precursor peptide, maturation protein, cyclodehydratase and dehydratase enzymes, the regulator, and the transporter. The precursor peptide is represented by the leader region (dark gray) and the core region (light gray). The predicted amino acid sequence is rich in cysteine and serine. The leader peptide is cleaved after maturation. (b) The chemical structure of the molecule is predicted from the amino acid sequence of the core peptide.

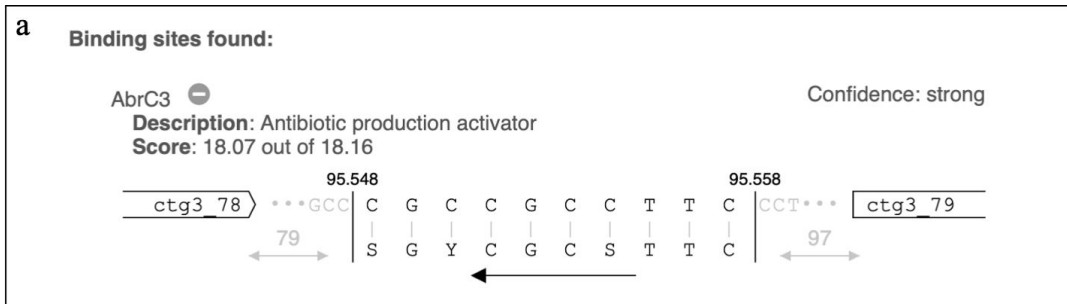

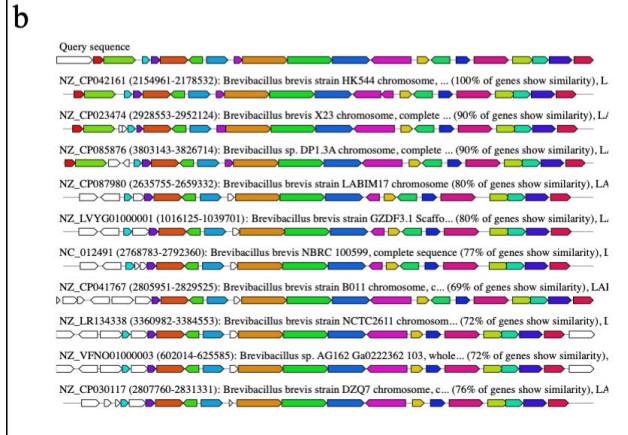

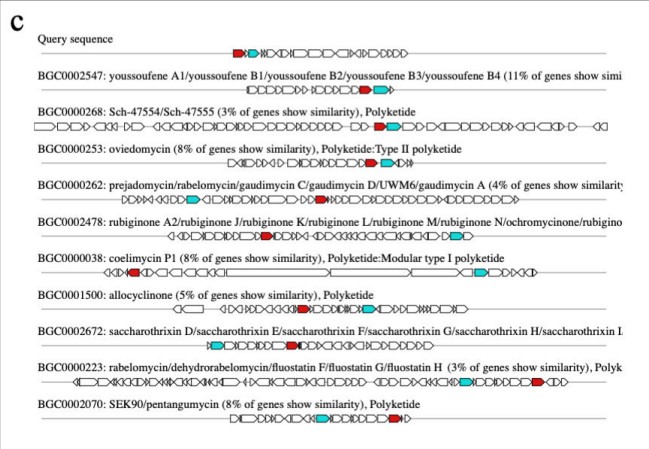

**FIG 9** (a) Binding sites to the transcription factor Abrc3. (b) ClusterBlast analysis with regions from the AntiSMASH database that are similar to the LAP BGC sequence. Genes marked with the same color are interrelated. Genes in white are unrelated. (c) Similar known gene clusters from MIBiG 3.1. Genes marked with the same color are interrelated. Genes in white are unrelated.

network cluster showed low similarity with known gene clusters from MIBiG (Fig. 9c). BGC 10 synthesizes a lanthipeptide of class II. It contains all genes necessary for the synthesis of regulatory elements and carries three genes for auto-resistance. Identification of antimicrobial resistance genes allows insights into self-defense mechanisms and suggests the production of specialized metabolites with biological activity (94). These BGCs are strong candidates for cloning and biosynthesis by heterologous expression to analyze the structure and functionality of the AMPs.

Metabolically, *Brevibacillus* sp. FIR094 showed a considerable node number synthesizing more metabolites when grown in R2A minimal medium, probably because this type of medium creates a nutritional stress that enhances secondary metabolite production (95). Despite the good results in metabolite production, most of the metabolome was not possible to be annotated, the poorest characterized metabolome of herein studied strains. BGCs for the synthesis of important antimicrobial compounds such as tyrocidine, gramicidin, and macrobrevin were detected in the genome, but we were unable to find these molecules in the metabolome. This may have occurred because of the metabolite extraction and detection methods that were not focused on large peptides or the tight regulation of their production, whose expression could be hampered under the laboratory conditions. The similarity network of *Brevibacillus* sp. FIR094 revealed metabolites with molecular weights of up to 1,200 Da. Tyrocidine and gramicidin have a molecular weight of 1,270.5 and 1,882.3 g/mol, respectively, a value above the threshold of detection.

## Conclusions

In this research, we evaluated the biosynthetic potential of three free-living bacterial strains isolated from soil samples collected in an Amazon Conservation Unit using

a multi-omics approach based on genome mining and untargeted metabolomics. Genome mining identified several BGCs for the synthesis of antimicrobial and antitumor compounds. However, a significant number of unidentified BGCs were found, corresponding to 45%, 88%, and 71% of the predicted clusters in *Streptomyces* sp. ACT015, *Rhodococcus* sp. ACT016, and *Brevibacillus* sp. FIR094including relevant biosynthetic classes, respectively, such as NRPS, PKS, and RiPPs, indicating the untapped potential of Amazonian microorganisms. The limited data in current databases and the presence of unidentified BGCs open doors for future research and offer excellent candidates for detailed characterization using gene cloning and heterologous expression techniques. Continued exploration of these strains could undoubtedly lead to significant advances and the discovery of new bioactive products.

The biosynthesis of secondary metabolites was modulated by the type of culture medium. Most of the acquired spectra were not annotated by the consulted databases (GNPS and UNPD-ISDB), revealing a significant gap of information in the available databases and/or an untapped chemical novelty of the assessed Amazonian strains, even for exhaustively studied taxa such as *Streptomyces* spp. The data reinforce that Amazonian microorganisms are an important source for the registry of natural compounds and a promising source for discovering new natural molecules.

Data integration in the metabologenomics approach allows for a more accurate search of new secondary metabolites. This type of approach is especially useful for samples that have been poorly explored. In this study, cases in which we could correlate the predicted BGC with its metabolite were rare, possibly due to the novelty of the BGCs and spectra detected. These results indicate a high genetic and metabolic variability in Amazonian microorganisms, which could be explored for the detection and production of metabolites of medical and biotechnological utility.

## ACKNOWLEDGMENTS

Authors would like to thank Pró-Reitoria de Pesquisa e Pós-Graduação (PROPESP) from Universidade Federal do Pará (UFPA) for the payment of the Article Processing Charges through the PAPQ 2024 Call.

This research was funded by Coordenação de Aperfeiçoamento de Pessoal de Nível Superior (CAPES) under the grant number 88887.465291/2019–00 and Instituto Serrapilheira under the grant number Serra-1709–19681. The authors thank the Brazilian Bioscience National Laboratory (LNBio-CNPEM) and the EngBio Laboratory (UFPA) for infrastructure and personal dedicated to the project.

A.C.M.F.O.: designed and executed the experiments. Performed the field expeditions. Wrote the manuscript and compiled the data for deposition; B.D.V.: designed the cultivation experiments and assisted on their execution. Assisted on the metabologenomics investigation; R.F.: designed and assisted on the metabolomics experiments; L.S.S.: assistance in field expeditions. Performed genome assemblies, enrichment, and comparative genomics analysis. Paper review and support in data management; A.A.O.V.: bioinformatics assistance in genome assemblies. Support in server access and data demands; D.A.G.: support in NGS data post-processing; A.S.: funding and coordination of the project; R.A.B.: Conceptualization, discussions based on experiments, funding, and co-coordination of the project; D.B.B.T.: Conceptualization, discussions based on experiments, funding, and co-coordination of the project; M.P.C.S.: Conceptualization, discussions based on experiments, funding, and co-coordination of the project. All authors contributed to the text and agreed with the final version of the paper.

## AUTHOR AFFILIATIONS

[1]Biological Engineering Laboratory, Innovation Space, Guamá Science and Technology Park, Belém, Pará, Brazil

[2]Center of Genomics and Systems Biology, Institute of Biological Sciences, Federal University of Pará, Belém, Pará, Brazil

³Brazilian Biosciences National Laboratory (LNBio), Brazilian Center for Research in Energy and Materials (CNPEM), Campinas, São Paulo, Brazil

## AUTHOR ORCIDs

Ana Carolina Favacho Miranda de Oliveira ⓘ http://orcid.org/0000-0003-4985-2354
Rafael Azevedo Baraúna ⓘ http://orcid.org/0000-0002-7837-7380
Daniela Barretto Barbosa Trivella ⓘ http://orcid.org/0000-0002-7505-2345

## AUTHOR CONTRIBUTIONS

Ana Carolina Favacho Miranda de Oliveira, Data curation, Formal analysis, Investigation, Methodology, Writing – original draft | Bruna Domingues Vieira, Formal analysis, Investigation, Methodology | Rafael de Felício, Formal analysis, Investigation, Methodology | Lucas da Silva e Silva, Formal analysis, Investigation, Methodology | Adonney Allan de Oliveira Veras, Data curation, Formal analysis, Methodology, Software, Validation | Diego Assis das Graças, Investigation, Validation, Writing – review and editing | Rafael Azevedo Baraúna, Conceptualization, Data curation, Formal analysis, Funding acquisition, Investigation, Methodology, Project administration, Writing – original draft, Writing – review and editing | Daniela Barretto Barbosa Trivella, Conceptualization, Data curation, Formal analysis, Funding acquisition, Investigation, Project administration, Resources, Supervision, Validation, Writing – review and editing | Maria Paula Cruz Schneider, Conceptualization, Data curation, Funding acquisition, Investigation, Project administration, Resources, Supervision, Validation, Writing – review and editing.

## DATA AVAILABILITY

Genome sequences were submitted to GenBank under the BioProject ID PRJNA1089999 and accession numbers SAMN40553816, SAMN40553817, and SAMN40553818. Metabolomics data can be found at: https://massive.ucsd.edu/, and accession numbers MSV000094460, MSV000094461 and MSV000094462.

## ETHICS APPROVAL

Sampling was authorized by environmental authorities, the Instituto de Desenvolvimento Florestal e da Biodiversidade do Estado do Pará – IDEFLOR-BIO (N° 2021/830676), and the Instituto Chico Mendes de Conservação da Biodiversidade – ICMBio through the online system SISBIO (N° 78518–1). Access to the Brazilian genetic heritage was also registered in the Sistema Nacional de Gestão do Patrimônio Genético e do Conhecimento Tradicional Associado – SisGen (N° A8DA4E2).

## ADDITIONAL FILES

The following material is available online.

### Supplemental Material

**Supplemental material (Spectrum00996-24-s0001.docx).** Fig. S1 to Fig. S3: Table S1.

### Open Peer Review

**PEER REVIEW HISTORY (review-history.pdf).** An accounting of the reviewer comments and feedback.

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
