## [Reviewer comments · Microbiology Spectrum]

Microbiology Spectrum

A metabologenomics approach reveals the unexplored biosynthetic potential of bacteria isolated from an Amazon Conservation Unit

Ana Carolina de Oliveira, Bruna Vieira, Rafael de Felício, Lucas Silva, Adonney Veras, Diego Graças, Artur Silva, Rafael Azevedo Baraúna, Daniela Barretto Barbosa Trivella, and Maria Paula Schneider

Corresponding Author(s): Rafael Azevedo Baraúna, Universidade Federal do Para

Review Timeline:

Submission Date:	April 18, 2024
Editorial Decision:	July 12, 2024
Revision Received:	October 15, 2024
Accepted:	November 5, 2024

Editor: Bernadette Connors

Reviewer(s): The reviewers have opted to remain anonymous.

Transaction Report:

DOI: <https://doi.org/10.1128/spectrum.00996-24>

Re: Spectrum00996-24 (A metabologenomics approach reveals the unexplored biosynthetic potential of bacteria isolated from pristine Amazonian soil)

Dear Prof. Rafael Azevedo Baraúna:

Thank you for the privilege of reviewing your work. Below you will find my comments, instructions from the Spectrum editorial office, and the reviewer comments.

Revision Guidelines

Sincerely,
Bernadette Connors
Editor
Microbiology Spectrum

Reviewer #2 (Comments for the Author):

In this study, the authors isolated bacterial strains from the Amazon and perform a process that they refer to as a "metabologenomics approach" to understand the potential for novel natural products discovery in this particular environment. The authors claim that there is vast unexplored chemical diversity from conserved regions in the Amazon using their approach to characterize three strains. However, this claim was never explicitly tested. It is not clear whether any three strains isolated

from any environment at random would not yield the same results.

In the study methodology, it is explained that the authors were looking for Actinomycetes and establish criteria for selecting strains (starting at L118). I found this confusing because one of the bacteria described in this study, *Brevibacillus* sp. FIR094, is not an Actinomycete. More explanation of this criteria, when it was or wasn't followed, and citations for why these are relevant criteria are needed here.

In the conclusion, there is discussion of the untapped potential of Amazonian microbes, however there is no quantitative analysis to demonstrate how under sampled this region is compared to others. In the final paragraph of the Conclusion, the authors mention the potential usefulness of pristine Amazonian soils (I'm assuming they are referring to the samples used here). However, they sampled in a park that was only established in 1993. I think it's worth characterizing these samples in a more realistic way (in other words, they are part of a conservation area, but likely have been impacted by anthropogenic activity at some point in their history). Otherwise, the authors are downplaying the potential in any environments and promoting bioprospecting pressure in areas that important for conservation.

While the description of these strains is thorough, the rationale for why these were sampled is not very strong. Further, it seems like the authors may not follow their own selection criteria (or it is written in a confusing manner).

Reviewer #3 (Comments for the Author):

This paper titled "A metabologenomics approach reveals the unexplored biosynthetic potential of bacteria isolated from pristine Amazonian soil" presents a comprehensive study on the biosynthetic potential of Gram-positive bacteria isolated from pristine Amazonian soils using a multi-omics approach. The authors effectively combined genome mining and untargeted metabolomics to identify numerous biosynthetic gene clusters (BGCs) and their corresponding metabolites. The findings reveal a significant number of BGCs related to unknown metabolites, underscoring the unexplored chemical diversity of Amazonian microorganisms. While the study is robust and provides valuable insights, improvements in abstract conciseness, methodology clarity, and a deeper discussion of results would enhance the overall impact. Overall, this paper contributes significantly to the field of microbial natural product discovery. My comments on the manuscript and supporting information document are detailed below.

General comments:

Abstract Clarity and Conciseness:

The abstract is lengthy and could be more concise. It should succinctly summarize the study's main findings and significance. Highlight the key discoveries, such as the identification of specific biosynthetic gene clusters (BGCs) and the types of metabolites produced. Provide a brief overview of the methods used without going into specifics.

Introduction Depth:

1. The introduction briefly mentions the importance of Amazonian biodiversity and the bioeconomy but lacks depth in discussing existing research on microbial communities and natural product discovery.

2. Clearly articulate why the chosen multi-omics approach is significant and how it addresses the limitations of previous methods. Emphasize the novelty of integrating genome mining with untargeted metabolomics in the context of Amazonian soil microbes

Methodology Details:

While the methodology is detailed, it could be structured more clearly. For instance, subsections with headings for each major part of the methodology (e.g., sampling, bacterial isolation, genome mining) would improve readability.

Statistical Analysis:

The paper lacks a detailed explanation of the statistical methods used to analyze the data. Including this information would strengthen the validity of the findings.

Discussion Section:

The discussion could be expanded to include more comparisons with previous studies and a deeper analysis of the implications of the findings.

Conclusion Depth:

The conclusion is brief and could be expanded to better highlight the broader implications of the study's findings and suggest future research directions.

Proofreading:

There are minor grammatical and typographical errors throughout the document. A thorough proofreading session is necessary to polish the paper.

Minor grammatical and typographical errors throughout the document

Abstract:

1. Line 24: "characterize the metabolic potential of in Gram-positive strains" - The word "in" should be removed.

2. Line 33: (62,5%)? Check.

3. Line 50: "ribosomal or non-ribosomal synthesis" - Should be hyphenated as "ribosomal- or non-ribosomal-synthesis."

4. Line 54: "plethora of applications, such as antibiotics for human health" - Should be "plethora of applications, including antibiotics for human health."

5. Line 78: "Same strategy was used" should be "The same strategy was used."
6. Line 97: "Bacterial isolation, taxonomic classification, and selection of strains" - This heading should be bold or formatted as a section heading for better readability.
7. Line 129: "to the three selected strain" should be "to the three selected strains."
8. Line 177: "followed by an MS/MS scan for the most intense ions in a 1 s cycle time, absolute threshold (per 1,000 sums) of 1,500 cts." - This sentence is fragmented and could be rephrased for clarity.
9. Line 207: "He strains ACT015, ACT016, and FIR094" - "He" should be "The."
10. Line 239: "were summarized in table 1 and were in accordance" - "table" should be capitalized.
11. Line 268: "from the same cluster, different types of metabolites (34)." "from the same cluster, producing different types of metabolites (34)."
12. Line 309: "were not annotated. When we applied restrictive parameters for spectra annotation" - "applied" should be "applied."
13. Line 327: "could not be made for another BGCs" - Should be "other BGCs."
14. Line 356: "The most of BGCs" - Should be "Most of the BGCs."
15. Line 397: "UNPD-ISDB annotated a grater number of spectra" - "grater" should be "greater."
16. Line 407: "these in silico tools to better describe the chemical potential" - "chemical potential" could be more specific as "biosynthetic potential."
17. Line 415: "genome size of Rhodococcus species varies between 4 and 10 Mbp" - "Mbp" should be "Mb."
18. Line 417: "genome size is related" - "size" should be "sizes."
19. Line 424: "genome size of the strain ACT016 isolated" - "isolated" is misplaced, making the sentence unclear.
20. Line 429: "Doroghazi and Metcalf (61) demonstrated that Rhodococcus isolates presented a greater proportion" - "presented" should be "present."
21. Line 441: "Rhodococcus sp. ACT016 showed higher production of metabolites in A1 and TSB media" - "higher" should be "a higher."
22. Line 450: "Some Rhodococcus species such as *R. opacus*, *R. jostii*, and *R. wratislaviensis*, are known to be oleaginous microorganisms" - The comma after "*wratislaviensis*" is unnecessary.
23. Line 457: "was composed by peptone digest" - "by" should be "of."
24. Line 485: "the genomic region is highly conserved in the genus" - "in" should be "within."
25. Line 493: "contains all genes necessary for the synthesis of regulatory elements and carries three genes for auto resistance" - "auto resistance" should be hyphenated as "auto-resistance."
26. Line 519: "Streptomyces sp. ACT015, Rhodococcus sp. ACT016, and Brevibacillus sp. FIR094, respectively, including relevant biosynthetic classes" - "respectively" should be moved to after "including relevant biosynthetic classes."
27. Line 531: "The biosynthesis of secondary metabolites was modulated by the type of culture media" - "media" should be "medium."
28. Line 537: "Amazonian microorganisms are, therefore, an important source" - The comma after "are" is unnecessary.
29. Line 549: "Authors thank the Brazilian Bioscience National Laboratory" - "Authors" should be "The authors."

This paper titled “**A metabologenomics approach reveals the unexplored biosynthetic potential of bacteria isolated from pristine Amazonian soil**” presents a comprehensive study on the biosynthetic potential of Gram-positive bacteria isolated from pristine Amazonian soils using a multi-omics approach. The authors effectively combined genome mining and untargeted metabolomics to identify numerous biosynthetic gene clusters (BGCs) and their corresponding metabolites. The findings reveal a significant number of BGCs related to unknown metabolites, underscoring the unexplored chemical diversity of Amazonian microorganisms. While the study is robust and provides valuable insights, improvements in abstract conciseness, methodology clarity, and a deeper discussion of results would enhance the overall impact. Overall, this paper contributes significantly to the field of microbial natural product discovery. My comments on the manuscript and supporting information document are detailed below.

General comments:

Abstract Clarity and Conciseness:

The abstract is lengthy and could be more concise. It should succinctly summarize the study's main findings and significance.

Highlight the key discoveries, such as the identification of specific biosynthetic gene clusters (BGCs) and the types of metabolites produced. Provide a brief overview of the methods used without going into specifics.

Introduction Depth:

1. The introduction briefly mentions the importance of Amazonian biodiversity and the bioeconomy but lacks depth in discussing existing research on microbial communities and natural product discovery.
2. Clearly articulate why the chosen multi-omics approach is significant and how it addresses the limitations of previous methods. Emphasize the novelty of integrating genome mining with untargeted metabolomics in the context of Amazonian soil microbes

Methodology Details:

While the methodology is detailed, it could be structured more clearly. For instance, subsections with headings for each major part of the methodology (e.g., sampling, bacterial isolation, genome mining) would improve readability.

Statistical Analysis:

The paper lacks a detailed explanation of the statistical methods used to analyze the data. Including this information would strengthen the validity of the findings.

Discussion Section:

The discussion could be expanded to include more comparisons with previous studies and a deeper analysis of the implications of the findings.

Conclusion Depth:

The conclusion is brief and could be expanded to better highlight the broader implications of the study's findings and suggest future research directions.

Proofreading:

There are minor grammatical and typographical errors throughout the document. A thorough proofreading session is necessary to polish the paper.

Minor grammatical and typographical errors throughout the document

Abstract:

1. **Line 24:** "characterize the metabolic potential of in Gram-positive strains" - The word "in" should be removed.
2. **Line 33:** (62,5%)? Check.
3. **Line 50:** "ribosomal or non-ribosomal synthesis" - Should be hyphenated as "ribosomal-or non-ribosomal-synthesis."
4. **Line 54:** "plethora of applications, such as antibiotics for human health" - Should be "plethora of applications, including antibiotics for human health."
5. **Line 78:** "Same strategy was used" should be "The same strategy was used."
6. **Line 97:** "Bacterial isolation, taxonomic classification, and selection of strains" - This heading should be bold or formatted as a section heading for better readability.
7. **Line 129:** "to the three selected strain" should be "to the three selected strains."
8. **Line 177:** "followed by an MS/MS scan for the most intense ions in a 1 s cycle time, absolute threshold (per 1,000 sums) of 1,500 cts." - This sentence is fragmented and could be rephrased for clarity.
9. **Line 207:** "He strains ACT015, ACT016, and FIR094" - "He" should be "The."
10. **Line 239:** "were summarized in table 1 and were in accordance" - "table" should be capitalized.

11. **Line 268:** "from the same cluster, different types of metabolites (34)." "from the same cluster, producing different types of metabolites (34)."
12. **Line 309:** "were not annotated. When we applied restrictive parameters for spectra annotation" - "applied" should be "applied."
13. **Line 327:** "could not be made for another BGCs" - Should be "other BGCs."
14. **Line 356:** "The most of BGCs" - Should be "Most of the BGCs."
15. **Line 397:** "UNPD-ISDB annotated a grater number of spectra" - "grater" should be "greater."
16. **Line 407:** "these in silico tools to better describe the chemical potential" - "chemical potential" could be more specific as "biosynthetic potential."
17. **Line 415:** "genome size of Rhodococcus species varies between 4 and 10 Mbp" - "Mbp" should be "Mb."
18. **Line 417:** "genome size is related" - "size" should be "sizes."
19. **Line 424:** "genome size of the strain ACT016 isolated" - "isolated" is misplaced, making the sentence unclear.
20. **Line 429:** "Doroghazi and Metcalf (61) demonstrated that Rhodococcus isolates presented a greater proportion" - "presented" should be "present."
21. **Line 441:** "Rhodococcus sp. ACT016 showed higher production of metabolites in A1 and TSB media" - "higher" should be "a higher."
22. **Line 450:** "Some Rhodococcus species such as *R. opacus*, *R. jostii*, and *R. wratislaviensis*, are known to be oleaginous microorganisms" - The comma after "*wratislaviensis*" is unnecessary.
23. **Line 457:** "was composed by peptone digest" - "by" should be "of."
24. **Line 485:** "the genomic region is highly conserved in the genus" - "in" should be "within."
25. **Line 493:** "contains all genes necessary for the synthesis of regulatory elements and carries three genes for auto resistance" - "auto resistance" should be hyphenated as "auto-resistance."
26. **Line 519:** "Streptomyces sp. ACT015, Rhodococcus sp. ACT016, and Brevibacillus sp. FIR094, respectively, including relevant biosynthetic classes" - "respectively" should be moved to after "including relevant biosynthetic classes."

27. **Line 531:** "The biosynthesis of secondary metabolites was modulated by the type of culture media" - "media" should be "medium."
28. **Line 537:** "Amazonian microorganisms are, therefore, an important source" - The comma after "are" is unnecessary.
29. **Line 549:** "Authors thank the Brazilian Bioscience National Laboratory" - "Authors" should be "The authors."

Universidade Federal do Pará
Instituto de Ciências Biológicas
Centro de Genômica e Biologia de Sistemas

October 10th, 2024

Response to Reviewers Letter

Dear Editor,

Below you will find the point-by-point answers for each issue raised by the reviewers. We would like to highlight that the line numbers described below refer to the document containing changes in traceable mode.

Reviewer #2:

- I. In this study, the authors isolated bacterial strains from the Amazon and perform a process that they refer to as a "metabologenomics approach" to understand the potential for novel natural products discovery in this particular environment. The authors claim that there is vast unexplored chemical diversity from conserved regions in the Amazon using their approach to characterize three strains. However, this claim was never explicitly tested. It is not clear whether any three strains isolated from any environment at random would not yield the same results.

R: We thank the reviewer for his comment and for highlighting the importance of validating the approach. However, we would like to emphasize that the term metabologenomics was not coined in our study. It is an older concept, coined by Doroghazi et al. (2014) to identify methodologies for prospecting biosynthetic gene clusters using genomics and metabolomics in an integrated manner. The statement about the vast unexplored chemical diversity refers only to the isolated bacterial species and not to the Amazon biome as a whole. This statement is based on the observation that much of the metabolic potential of the bacterial strains has not been identified by existing database platforms. This suggests a rich metabolic diversity that has not yet been catalogued. Although we selected strains that are commonly used in studies of prospection of bioactive compounds, our results highlight the uniqueness and untapped potential of Amazonian strains. We have tried to make this clearer in this revised version of the manuscript and hope that the changes meet your expectations.

Doroghazi JR, Albright JC, Goering AW et al. (2014) A roadmap for natural product discovery based on large-scale genomics and metabolomics. Nature Chemical Biology, 10(11):963-968.

- II. In the study methodology, it is explained that the authors were looking for Actinomycetes and establish criteria for selecting strains (starting at L118). I found this confusing because one of the bacteria described in this study, *Brevibacillus* sp.

FIR094, is not an Actinomycete. More explanation of this criteria, when it was or wasn't followed, and citations for why these are relevant criteria are needed here.

R: We regret that the information was not complete in the initial manuscript submission. This issue has been rectified in the revised document, as detailed in lines 42-44 and 154-163.

III. In the conclusion, there is discussion of the untapped potential of Amazonian microbes, however there is no quantitative analysis to demonstrate how under sampled this region is compared to others. In the final paragraph of the Conclusion, the authors mention the potential usefulness of pristine Amazonian soils (I'm assuming they are referring to the samples used here). However, they sampled in a park that was only established in 1993. I think it's worth characterizing these samples in a more realistic way (in other words, they are part of a conservation area, but likely have been impacted by anthropogenic activity at some point in their history). Otherwise, the authors are downplaying the potential in any environments and promoting bioprospecting pressure in areas that important for conservation.

R: We recognize the importance of providing a quantitative analysis that demonstrates how under-sampled the Amazon region is compared to other regions. During the analysis of our data, we observed that the existing scientific literature is deficient in studies based on environmental samples from the Amazon. The paucity of studies in this field made it challenging to compare our findings with other research conducted in the region, emphasizing how much remains to be explored in the Amazon. In this revised version, we have attempted to clarify that the term "vast untapped biosynthetic potential" refers specifically to the microorganisms under study and not to the biome as a whole. Furthermore, the term "pristine soils" has been replaced with "Amazon Conservation Unit soil," as recommended by the reviewer (changes were made to the title and lines 249, 618, 639).

IV. While the description of these strains is thorough, the rationale for why these were sampled is not very strong. Further, it seems like the authors may not follow their own selection criteria (or it is written in a confusing manner).

R: We offer our sincere apologies if the description was unclear and would be pleased to provide a more detailed explanation. The criteria employed to select strains with biosynthetic potential were as follows (lines 154-163):

1. First filter was the taxonomic classification: The strains selected belonged to the Actinobacteria and Firmicutes phyla, which are widely recognized for producing a variety of secondary metabolites with biotechnological potential. These phyla have been extensively documented in the literature for their role in the biosynthesis of bioactive compounds.

2. Second filter was the inhibitory activity: The inhibitory activity of the strains was evaluated against control strains using the cross-streak method. This method allowed for the identification of strains capable of inhibiting the growth of other microorganisms, suggesting the production of substances with antibiotic potential or other biological activities.

Reviewer #3:

Abstract Clarity and Conciseness.

The abstract is lengthy and could be more concise. It should succinctly summarize the study's main findings and significance.

Highlight the key discoveries, such as the identification of specific biosynthetic gene clusters (BGCs) and the types of metabolites produced. Provide a brief overview of the methods used without going into specifics.

R: Thank you for your valuable suggestions. As requested, the abstract has been revised to be more concise and clear. The main findings have been highlighted, including the identification of biosynthetic gene clusters (BGCs) and the types of metabolites produced. Additionally, an overview of the methods used has been provided, without excessive detail. We hope these changes fulfil your expectations (lines 41-53).

Introduction Depth:

1. The introduction briefly mentions the importance of Amazonian biodiversity and the bioeconomy but lacks depth in discussing existing research on microbial communities and natural product discovery.

2. Clearly articulate why the chosen multi-omics approach is significant and how it addresses the limitations of previous methods. Emphasize the novelty of integrating genome mining with untargeted metabolomics in the context of Amazonian soil microbes

R: A paragraph has been included in the text which discusses the bioprospecting of natural products from microbial communities in the Amazon. This paragraph highlights the promising potential for new molecules to be discovered, including volatile organic compounds with antimicrobial activities and β -glucosidase inhibitors produced by cyanobacteria. In addition, a new biosurfactant from *Pseudomonas aeruginosa* with antiviral, antitumour and antimicrobial activities was described (lines 63-68). Furthermore, we seek to emphasise the advances in untargeted metabolomics, highlighting how more sensitive mass spectrometry (MS/MS) systems have enabled deeper analyses of metabolic components (lines 86-89).

Methodology Details:

While the methodology is detailed, it could be structured more clearly. For instance, subsections with headings for each major part of the methodology (e.g., sampling, bacterial isolation, genome mining) would improve readability.

R: In order to improve the clarity and structure of the text, each major part of the methodology has been described in a subsection such as sampling, bacterial isolation, genomic mining, etc. These modifications were designed to facilitate the comprehension of the methodology employed in the study.

Statistical Analysis:

The paper lacks a detailed explanation of the statistical methods used to analyze the data. Including this information would strengthen the validity of the findings.

R: The present study is exploratory in nature, and thus, no causal relationship can be inferred. Consequently, statistical analysis is not a prerequisite for supporting the study's conclusions. In instances where statistical analysis is conducted, such as in the prospecting of BGCs in genomes, the software utilized for BGC prediction performs the necessary statistical analysis, as indicated in the methodology. We hope this explanation helps to clarify this issue.

Discussion Section and Conclusion Depth:

The discussion could be expanded to include more comparisons with previous studies and a deeper analysis of the implications of the findings.

R: In order to fulfil your recommendations in the discussion and conclusion section, we have sought to expand the text by including more detailed comparisons with previous studies, with a particular focus on the genus *Streptomyces*, which has been extensively explored in comparison to the genera *Rhodococcus* and *Brevibacillus*. Moreover, in the conclusion, we have included a paragraph that emphasizes the deficiencies in current databases and the existence of unidentified BGCs. This reinforces the assertion that these areas present promising opportunities for future investigation. We hope these changes meet your expectations and are satisfactory (lines 381-385, 388-401, 407-422, 436-440, 455-473).

Proofreading:

There are minor grammatical and typographical errors throughout the document. A thorough proofreading session is necessary to polish the paper.

R: We are thankful for your detailed suggestions regarding grammatical and typographical errors. We have implemented all the requested changes, as follows:

1. Line 24: "characterize the metabolic potential of in Gram-positive strains" - The word "in" should be removed.

R: The abstract was revised. The word "in" was removed.

2. Line 33: (62,5%)? Check.

R: The abstract was revised. The percentage was checked.

3. Line 50: "ribosomal or non-ribosomal synthesis" - Should be hyphenated as "ribosomal- or non-ribosomal-synthesis."

R: The correction was done as suggested (line 69).

4. Line 54: "plethora of applications, such as antibiotics for human health" - Should be "plethora of applications, including antibiotics for human health."

R: The correction was done as suggested (lines 73-74).

5. Line 78: "Same strategy was used" should be "The same strategy was used."

R: The correction was done as suggested (line 96).

6. Line 97: "Bacterial isolation, taxonomic classification, and selection of strains" - This heading should be bold or formatted as a section heading for better readability.

R: All subsections were highlighted in bold.

7. Line 129: "to the three selected strain" should be "to the three selected strains."

R: The correction was done as suggested (line 167).

8. Line 177: "followed by an MS/MS scan for the most intense ions in a 1 s cycle time, absolute threshold (per 1,000 sums) of 1,500 cts." - This sentence is fragmented and could be rephrased for clarity.

R: This sentence was revised as suggested: “This was followed by an MS/MS scan for the most intense ions, with a cycle time of 1 second and an absolute threshold (per 1,000 sums) of 1,500 cts.” (lines 218-220).

9. Line 207: "He strains ACT015, ACT016, and FIR094" - "He" should be "The."

R: Thank you! The correction was done as suggested (line 250).

10. Line 239: "were summarized in table 1 and were in accordance" - "table" should be capitalized.

R: The correction was done as suggested (line 257).

11. Line 268: "from the same cluster, different types of metabolites (34)." "from the same cluster, producing different types of metabolites (34)."

R: The correction was done as suggested (lines 297-298).

12. Line 309: "were not annotated. When we appled restrictive parameters for spectra annotation" - "appled" should be "applied."

R: The correction was done as suggested (line 337).

13. Line 327: "could not be made for another BGCs" - Should be "other BGCs."

R: The correction was done as suggested (line 366).

14. Line 356: "The most of BGCs" - Should be "Most of the BGCs."

R: The correction was done as suggested (line 427).

15. Line 397: "UNPD-ISDB annotated a grater number of spectra" - "grater" should be "greater."

R: The correction was done as suggested (line 492).

16. Line 407: "these in silico tools to better describe the chemical potential" - "chemical potential" could be more specific as "biosynthetic potential."

R: The correction was done as suggested (line 510).

17. Line 415: "genome size of Rhodococcus species varies between 4 and 10 Mbp" - "Mbp" should be "Mb."

R: The correction was done as suggested (line 514).

18. Line 417: "genome size is related" - "size" should be "sizes."

R: The correction was done as suggested (line 515).

19. Line 424: "genome size of the strain ACT016 isolated" - "isolated" is misplaced, making the sentence unclear.

R: The sentence was revised and changed to “genome size of the strain ACT016 was 5.8 Mb” (line 521).

20. Line 429: "Doroghazi and Metcalf (61) demonstrated that Rhodococcus isolates presented a greater proportion" - "presented" should be "present."

R: The correction was done as suggested (line 526).

21. Line 441: "Rhodococcus sp. ACT016 showed higher production of metabolites in A1 and TSB media" - "higher" should be "a higher."

R: The correction was done as suggested (line 537).

22. Line 450: "Some Rhodococcus species such as *R. opacus*, *R. jostii*, and *R. wratislaviensis*, are known to be oleaginous microorganisms" - The comma after "*wratislaviensis*" is unnecessary.

R: The correction was done as suggested (line 546).

23. Line 457: "was composed by peptone digest" - "by" should be "of."

R: The correction was done as suggested (line 581).

24. Line 485: "the genomic region is highly conserved in the genus" - "in" should be "within."

R: The correction was done as suggested (line 593).

25. Line 493: "contains all genes necessary for the synthesis of regulatory elements and carries three genes for auto resistance" - "auto resistance" should be hyphenated as "auto-resistance."

R: The correction was done as suggested (line 596).

26. Line 519: "Streptomyces sp. ACT015, Rhodococcus sp. ACT016, and Brevibacillus sp. FIR094, respectively, including relevant biosynthetic classes" - "respectively" should be moved to after "including relevant biosynthetic classes."

R: The correction was done as suggested (line 624)

27. Line 531: "The biosynthesis of secondary metabolites was modulated by the type of culture media" - "media" should be "medium."

R: The correction was done as suggested (line 631).

28. Line 537: "Amazonian microorganisms are, therefore, an important source" - The comma after "are" is unnecessary.

R: The correction was done as suggested (line 635).

29. Line 549: "Authors thank the Brazilian Bioscience National Laboratory" - "Authors" should be "The authors."

R: The correction was done as suggested (line 650).

Re: Spectrum00996-24R1 (A metabologenomics approach reveals the unexplored biosynthetic potential of bacteria isolated from an Amazon Conservation Unit)

Dear Prof. Rafael Azevedo Baraúna:

Your manuscript has been accepted, and I am forwarding it to the ASM production staff for publication. Your paper will first be checked to make sure all elements meet the technical requirements. ASM staff will contact you if anything needs to be revised before copyediting and production can begin. Otherwise, you will be notified when your proofs are ready to be viewed.

Sincerely,
Bernadette Connors
Editor
Microbiology Spectrum

Reviewer #2 (Comments for the Author):

Thank you for the changes based on the previous comments. Please make the line correction as seen below:

L 380: "Clique ou toque aqui para inserir o texto.." -> this was left in the body of the text and should be removed.